# Ciliary tip actin dynamics regulate photoreceptor outer segment integrity

Roly Megaw [1,2] ✉, Abigail Moye[3], Zhixian Zhang[3], Fay Newton [1], Fraser McPhie[1], Laura C. Murphy[1], Lisa McKie[1], Feng He[3], Melissa K. Jungnickel[1], Alex von Kriegsheim [4], Peter A. Tennant [1], Chloe Brotherton [1], Christine Gurniak[5], Alecia K. Gross[6], Laura M. Machesky [7,8], Theodore G. Wensel [3] & Pleasantine Mill [1]

As signalling organelles, cilia regulate their G protein-coupled receptor content by ectocytosis, a process requiring localised actin dynamics to alter membrane shape. Photoreceptor outer segments comprise an expanse of folded membranes (discs) at the tip of highly-specialised connecting cilia, into which photosensitive GPCRs are concentrated. Discs are shed and remade daily. Defects in this process, due to mutations, cause retinitis pigmentosa (RP). Whilst fundamental for vision, the mechanism of photoreceptor disc generation is poorly understood. Here, we show membrane deformation required for disc genesis is driven by dynamic actin changes in a process akin to ectocytosis. We show RPGR, a leading RP gene, regulates actin-binding protein activity central to this process. Actin dynamics, required for disc formation, are perturbed in *Rpgr* mouse models, leading to aborted membrane shedding as ectosome-like vesicles, photoreceptor death and visual loss. Actin manipulation partially rescues this, suggesting the pathway could be targeted therapeutically. These findings help define how actin-mediated dynamics control outer segment turnover.

Most mammalian cells assemble a primary cilium; a microtubule-based structure that protrudes from the cell body and functions as a sensory organelle by detecting changes in the extracellular environment and initiating signalling[1]. Cilia dysfunction, due to pathogenic mutations in critical genes, leads to a spectrum of human diseases termed the ciliopathies, which comprise isolated or multisystem disorders of the brain, lung, kidney and eye, amongst others[1]. Thus, tight control of cilia signalling is crucial for human health.

Cilia function is optimised by compartmentalising the initiators of signalling cascades, such as G protein-coupled receptors (GPCRs), in its membrane. This is achieved by high-volume trafficking to the cilia,

but more recently it has been shown that dynamic membrane changes at the ciliary tip can regulate GPCR concentration within the cilium in a process termed ectocytosis, which involves the shedding of cilia-membrane-derived vesicles into the extracellular space[2,3]. Ectosome formation is facilitated by local changes in the actin cytoskeleton to initiate the membrane deformation required to form these structures that will be subsequently pinched off and shed. How important this biological process is across cell types and in the context of human health remains unclear.

The photoreceptor contains one of the most highly specialised primary cilia- the connecting cilia, CC- that have evolved to optimise

[1]MRC Human Genetics Unit, MRC Institute of Genetics & Cancer, University of Edinburgh, Western General Hospital, Edinburgh EH4 2XU, UK. [2]Princess Alexandra Eye Pavilion, NHS Lothian, Edinburgh EH3 9HA, UK. [3]Verna and Marrs McLean Department of Biochemistry and Molecular Pharmacology, Baylor College of Medicine, Houston, TX 77030, USA. [4]Edinburgh Cancer Research United Kingdom Centre, Institute of Genetics and Cancer, University of Edinburgh, Edinburgh EH4 2XU, UK. [5]Institute fur Genetik, Universitat Bonn, Karlrobert-Kreiten-Strasse, 53115 Bonn, Germany. [6]University of Alabama at Birmingham, 2nd Ave South, Birmingham, AL 35294, USA. [7]CRUK Scotland Institute, Switchback Road, Bearsden, Glasgow G61 1BD, UK. [8]Department of Biochemistry, University of Cambridge, Cambridge CB1 7UY, UK. ✉e-mail: roly.megaw@ed.ac.uk

our visual processing capabilities by compartmentalising its photo-sensitive GPCRs within hundreds of disc-like membranous processes that stack on top of each other at the distal end of the cilia to form the cell's outer segment (OS)[4,5]. To enable recycling of its contents, the photoreceptor OS is completely renewed every 7 to 10 days[6], with distal discs shed for phagocytosis by the underlying retinal pigment epithelium (RPE)[7]. A fine balance, with continuous birth of photoreceptor discs to replace the shed OS material, is critical to support vision. The mechanism that drives the ciliary membrane remodelling required for disc formation is yet to be fully determined, but evidence is mounting that it is an actin-driven process[8-11] and it has been speculated that the process has evolved as a form of ectocytosis[9-11]. Failure to renew photoreceptor discs has been implicated in retinitis pigmentosa (RP)[10], a heterogenous group of inherited retinal dystrophies affecting 1 in 3000 people[12] that cause blindness. Patients present with night blindness and progressive constriction of their visual fields, prior to loss of central vision, as their photoreceptors degenerate.

Here, using cryo-electron tomography (cryoET) and mouse disease models, we provide evidence supporting a model whereby the membrane deformation required for photoreceptor disc formation is an actin-driven process akin to ectocytosis. Further, we show that the retinitis pigmentosa GTPase regulator (RPGR) protein, mutations in which cause 15% of RP[13], functions to bind the actin-severing protein cofilin in the distal photoreceptor cilia, regulating its activity. *RPGR* mutations compromise cofilin activity, resulting in lengthened actin bundles in the newly forming disc. As a result, compromised discs are shed as ectosome-like vesicles, resulting in outer segment abnormalities, retinal stress, photoreceptor degeneration and loss of vision. We conclude, therefore, that highly regulated actin control in the nascent photoreceptor disc controls disc integrity in the same manner as ectosome formation and that *Rpgr* plays a crucial role in the process.

## Results

### Disc formation is an active, actin-driven process

There is growing evidence that disc formation is actin-dependent[11,14-16]. It has not been definitively shown, however, if this is due to an active process, whereby progressive nucleation of actin microfilaments (akin to lamellipodia formation)[17] mechanically deform the membrane.

Alternatively, it could be a passive process (such as blebbing), whereby loss of adhesion between the membrane and underlying cortical actin allows the hydrostatic pressure within a cell to deform the membrane[18]. To distinguish between these possibilities in situ at nanoscale, we used cryoET[19]. Flash freezing isolated mouse rod photoreceptor outer segments (ROS) allows visualisation of 3-dimensional architecture at an ultrastructural level by creating 3-dimensional maps from a tilt series of electron tomograph images (Supplementary Fig. 1). Although membranes can break and reseal during the isolation process, internal structures such as filaments are well preserved[19,20]. To best distinguish between an active and a passive process, our annotation focussed on the unflattened, nascent disc emerging from the base of the CC. Three-dimensional reconstructions showed microfilaments extending from the distal CC into the nascent disc (Fig. 1a–c, Supplementary Movies 1–3). Sub-tomogram averaging of these microfilaments revealed similarities to filamentous actin (F-actin) published structure of 166.67° twists per molecule and a 27.8 Å rise per molecule (Fig. 1d–f)[21]. We thus conclude that dynamic actin changes occur within the photoreceptor CC, supporting a role for actin in disc genesis.

### RPGR is required for photoreceptor outer segment maintenance

Mutations in *RPGR* account for 70–90% of X-linked RP (XLRP) and 10-15% of all RP and result in a severe form of disease[13]. *RPGR* is alternatively spliced, with a retinal specific isoform containing a repetitive, disordered C-terminal domain of unknown function (conventionally termed 'ORF15') that is a mutational hotspot for disease[22]. RPGR localises to the photoreceptor CC and its loss results in perturbation of CC actin regulation, with subsequent photoreceptor degeneration[9,23-25]. To determine if RPGR plays a role in actin-driven disc genesis, we generated novel *Rpgr*-mutant mice harbouring humanised disease-causing mutations.

N terminal *RPGR* mutations are associated with more severe human disease than those in the C terminal ORF15 domain[26]. However, mouse genetic background has been shown to influence disease severity[27]. To determine if the murine disease correlated with that of human, and was therefore an appropriate model for this work, two novel mutations were generated in the same strain of mice, each replicating human pathogenic mutations (see Methods and Supplementary Fig. 2). One line harboured a frameshift mutation in *Rpgr's*

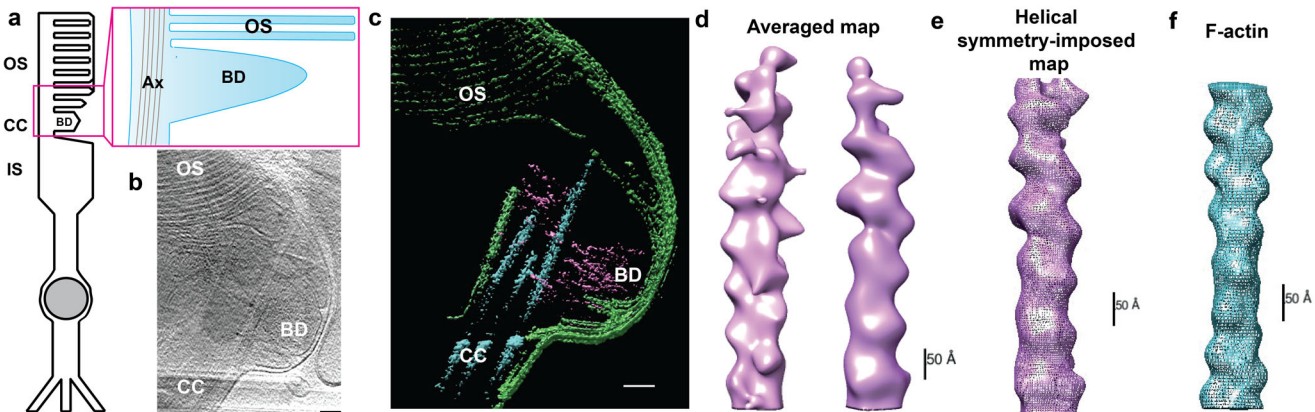

**Fig. 1 | Photoreceptor disc formation is an actin-driven process. a** Schematic of a rod photoreceptor depicts the cell's inner segment (IS) with the highly modified connecting cilium (CC) extending from its apex, containing a microtubule-based axoneme (Ax). From here an expanse of folded membrane extends, forming the outer segment (OS) discs. Basal discs (BD) are continually added at its base. **b** A projection from a tomogram of the basal disc of a wild-type photoreceptor. **c** A slice through a tomographic reconstruction, segmented to highlight the ciliary and disc membranes (green), microtubule-based axoneme (cyan) and microfilaments (purple). Microfilaments extend from the connecting cilium into the basal disc. **d** Front and back views of a subtomogram averaged map of microfilaments extending into the BD, with helical symmetry-imposed map (**e**). **f** Previously published structure of F-actin (EMBD-6448), highlighting 166.67 degree twist per molecule and 27.8 Å rise per molecule. (Scale bars; **b**, **c** = 100 nm; **d**–**f** = 50 Å) (Similar results were seen in 6 independent experiments, each pooling results from 9 to 32 eyes (8–16 mice) and 4–12 tomograms).

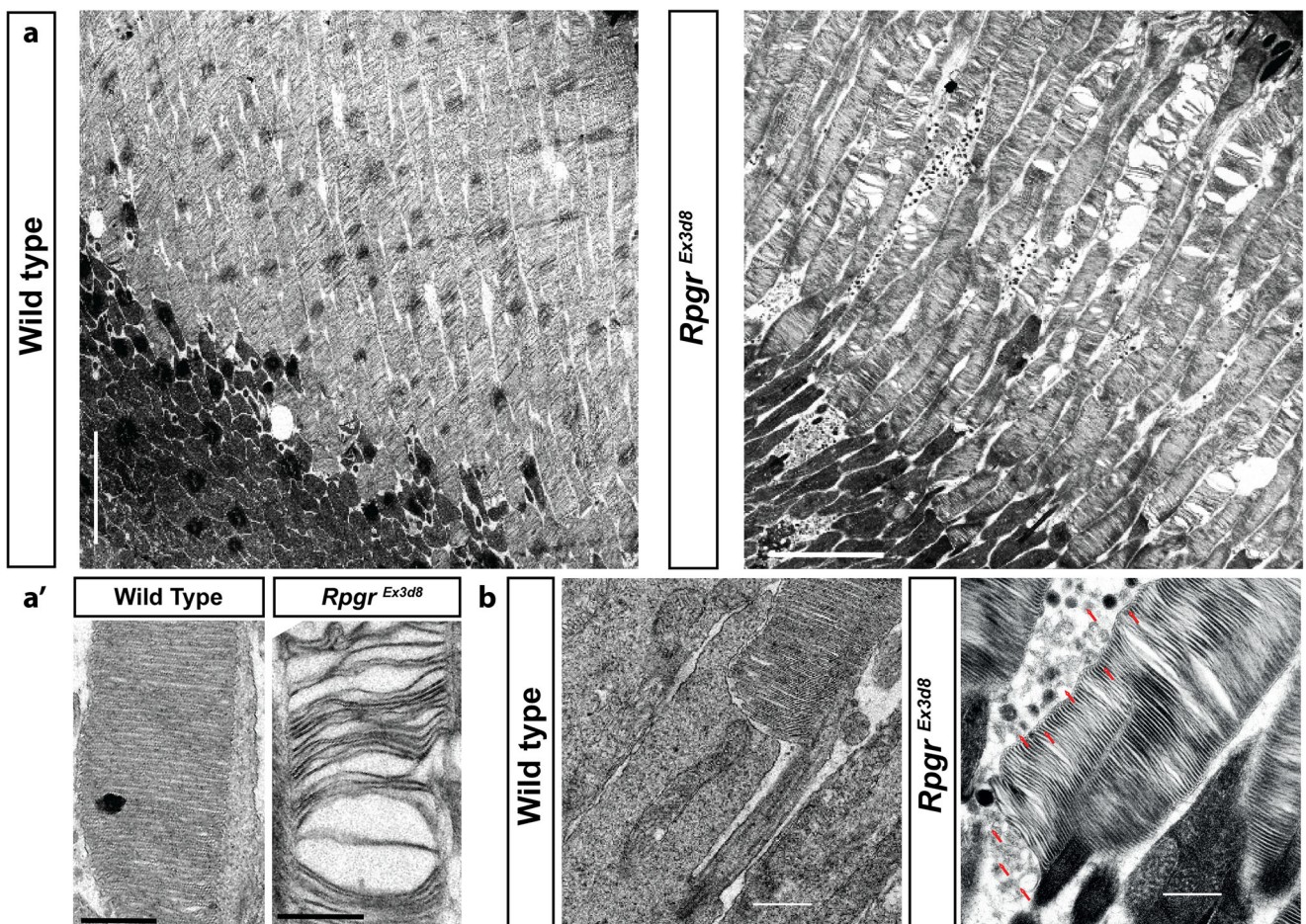

**Fig. 2 | Mutations in *Rpgr* lead to early structural compromise of photoreceptor outer segments (OS). a** Transmission electron micrograph of 6 week old wild-type photoreceptors show OS composed of compacted discs extending to the underlying retinal pigment epithelium (RPE) (left panel). Disc compaction is compromised in age-matched mutant photoreceptors, with discs appearing split and spaced out (right panel). **a'** Higher magnification examples. **b** Examination of basal discs at OS base shows compacted discs but an accumulation of shed vesicles (red arrows) in *Rpgr^Ex3d8^* mice. (Scale bars; **a** = 5 μm; **b** = 0.5 μm). (Similar results seen in >10 experiments repeated independently).

N-terminus (Supplementary Fig. 2a; hereby referred to as *Rpgr^Ex3d8^*), while the other harboured a truncation mutation in the C-terminal repetitive domain of the retina-specific isoform (Supplementary Fig. 2a; hereby referred to as *Rpgr^ORFd5^*).

Fundoscopy revealed retinal degeneration, characterised by punctate lesions at the posterior pole (Supplementary Figs. 3a, 4a) whilst fundus blue light autofluorescence showed accumulation of autofluorescent material within the retinas of both models (Supplementary Figs. 3b, 4b), a clinical biomarker for photoreceptor disease. Electroretinograms (ERGs) that measure retinal photoreceptor function were recorded as mice aged (Supplementary Figs. 3f, 5). The dark-adapted scotopic ERG reflects primarily rod photoreceptor function at lower stimulus intensities, whereas the light-adapted photopic ERG reflects cone photoreceptor function. Both models developed significant loss of their scotopic ERG a-wave amplitude (representing rod photoreceptor dysfunction) at 18 months (Supplementary Figs. 3f and 5). Cone photoreceptor function (photopic ERG a-wave amplitude) in the *Rpgr^ORFd5^ mouse* remained comparable to wild type but was significantly reduced from 6 months of age in the *Rpgr^Ex3d8^* mutant (Supplementary Fig. 5). Optical coherence tomography showed loss of the outer nuclear (photoreceptor) layer at 18 months (Supplementary Figs. 3c and 4c), which was supported by conventional histology in both models but was more severe in *Rpgr^Ex3d8^* (Supplementary Figs. 3d, e and 4d, e). In summary, loss of all *Rpgr* isoforms leads to more severe disease than loss of the retinal specific isoform alone, similar to human patient mutations. The *Rpgr^Ex3d8^ mouse* was brought forward for further studies.

We have previously shown *RPGR* mutations perturb actin regulation in both mouse and induced pluripotent stem cell-derived human photoreceptors[9]. Further, *RPGR* mutations result in loss of the retinal ellipsoid and interdigitating zones of XLRP patients on optical coherence tomography imaging (Supplementary Fig. 6); signifying OS loss. We therefore assessed the outer segments of our *Rpgr*-mutant mouse lines using transmission electron microscopy (TEM). At 6 weeks of age, long before the mice show signs of retinal stress (Supplementary Fig. 3g) and over 12 months before they undergo retinal degeneration (Supplementary Figs. 3d, e and 4d, e), mutant outer segments display a 'spaced disc' phenotype. Their discs have lost compaction and appear spaced out (Fig. 2a). Further, mutant outer segments are significantly shorter from base to RPE (Fig. 3a, b. Supplementary Fig. 7d) than wild type, suggesting *Rpgr^Ex3d8^* outer segments contain fewer discs. This conclusion is supported by proteomic data of 6 week mouse isolated rod outer segments and 3 month isolated retinas, which show *Rpgr^Ex3d8^* retinas contain lower amounts of photoreceptor disc component (PRCD, progressive cone-rod dystrophy) (Fig. 3c, Supplementary Data 1 and 2), a 6 kDa membrane protein that localises exclusively to photoreceptor discs and segregates to the outer disc rim[28–30]. Whilst no differences were observed on proteomics of other outer segment proteins (eg peripherin2, ABCA4), reduced PRCD in *Rpgr^Ex3d8^* retinas was confirmed on immunoblotting (Fig. 3d, e). Basal discs at the base

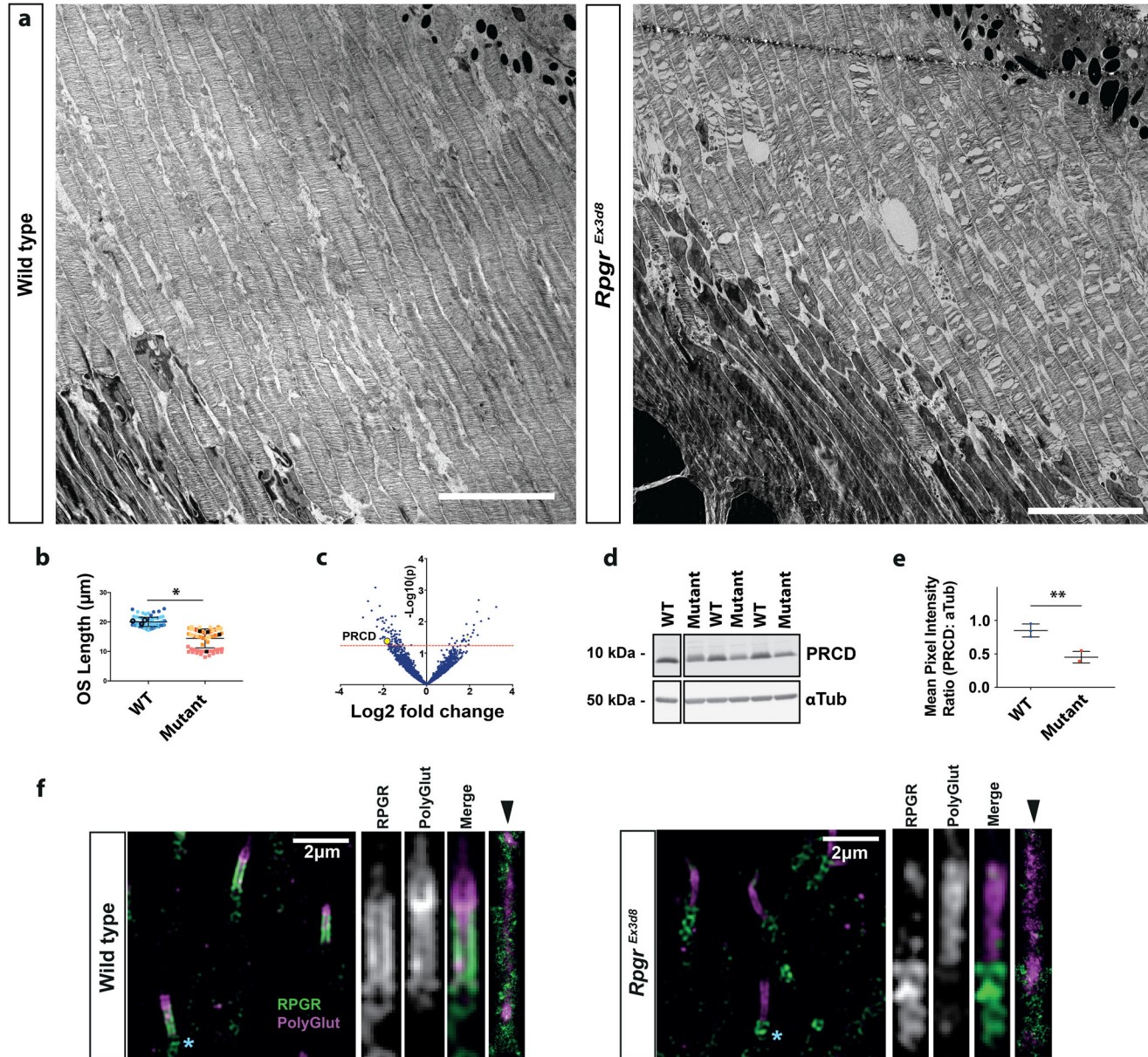

**Fig. 3 | Loss of *Rpgr* leads to shortened photoreceptor outer segments (OS).**
**a** Transmission electron micrograph of 6 week wild type and *Rpgr^Ex3d8* photo-receptors show OS are shorter in *Rpgr^Ex3d8* mice. **b** Quantification of OS length. Different colours represent measurements from individual mice. Mean OS length measurement of each experimental animal denoted by black circles (WT) and black squares (mutant); $N = 3$ animals per genotype; *, $p = 0.0405$; two-tailed, unpaired t-test; error bars denote SEM. **c** Mass spectrometry comparing protein composition of wild type versus *Rpgr^Ex3d8* 3 month old retinas shows reduced expression of the outer segment protein PRCD in mutant mice. Red line denotes cut-off p value for significance (two-tailed, unpaired t-test, not corrected for multiple hypothesis testing). **d** Immunoblotting of whole retina lysates confirms reduced PRCD in mutant mice, in keeping with reduced outer segment lengths. **e** Quantification of intensity of PRCD relative to loading control α-tubulin; $N = 3$ animals per group;

*$p = 0.0062$; two-tailed, unpaired t-test). Data presented as mean values +/− SEM. **f** Left panel: Localisation of RPGR's retinal-specific isoform throughout the length of the photoreceptor connecting cilium, as evidenced by SIM imaging, showing co-localisation with polyglutamylated tubulin in wild type photoreceptors, which extends the length of the connecting cilium and into the OS[63]. (Similar results seen in 4 independent experiments using separate animals). RPGR localizes to the ciliary membrane by STORM imaging (rightmost panel, arrowhead; similar results seen in 3 independent experiments). Right panel: *Rpgr^Ex3d8* photoreceptors show loss of RPGR staining at the connecting cilium. (NB. green staining below polyglutamyla-tion labelling (see blue *) represents non-specific centrosomal staining; a common occurrence with rabbit polyclonal antibodies) (Scale bars: **a** = 5 μm; **f** = 2 μm). Source data provided as a Source Data file.

of the *Rpgr^Ex3d8* outer segment are able to form (Fig. 2b), but disc for-mation appears perturbed, as *Rpgr^Ex3d8* mutants accumulated high levels of vesicles in the extracellular space at the base of outer seg-ments (Fig. 2b).

Vesicles of similar morphology were recently determined to be ciliary ectosomes in a RDS/peripherin mutant mouse (*rds^-/-*), which displays defective disc formation[10]. RPGR localises to the site of disc

formation in the distal CC (Fig. 3f) and could therefore play a role in disc genesis. Abnormal ciliary membrane deformation, with compro-mised discs instead shed as ectosomes, could slow the rate of com-pleted disc formation and account for this 'spaced disc' phenotype. We conclude that perturbation of RPGR compromises outer segment maintenance, leading to shortened, 'spaced disc' outer segments and vesicle shedding.

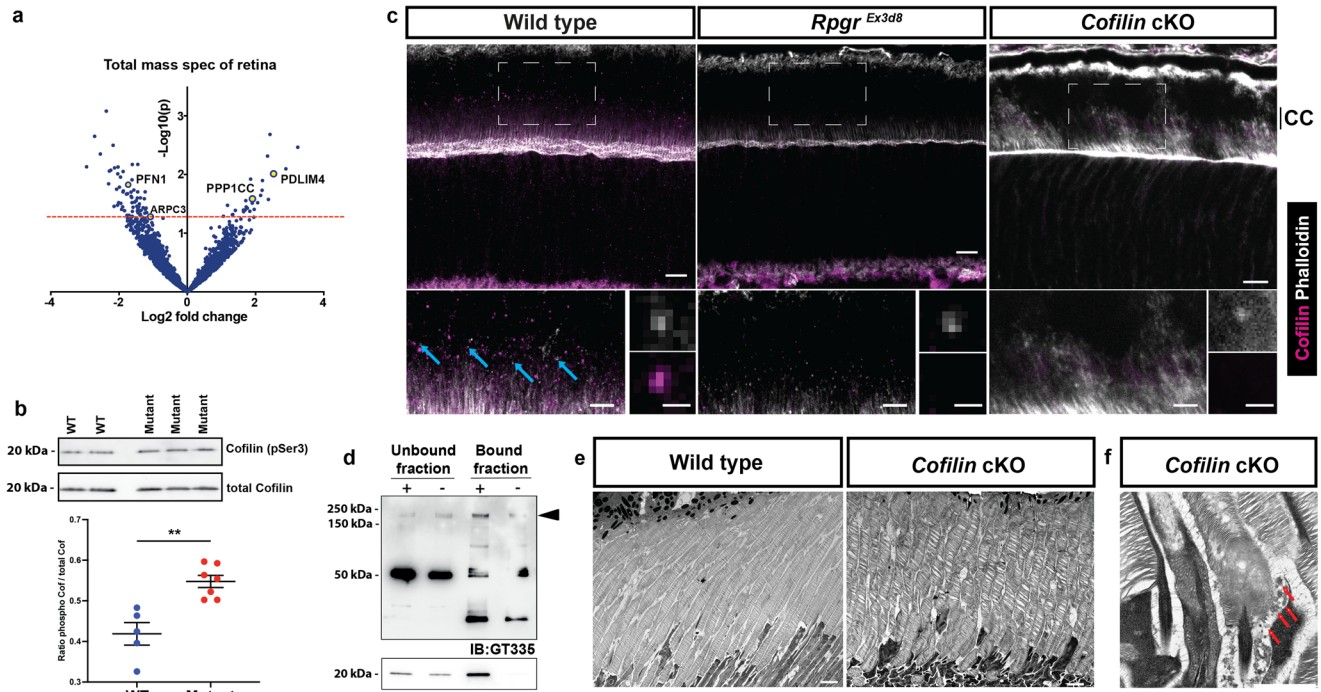

**Fig. 4 | RPGR binds and regulates activity of the actin-severing protein cofilin.**
**a** Mass spectrometry analysis of *Rpgr*$^{Ex3d8}$ retina shows dysregulation of actin regulators (PFN1, PDLIM4, PPP1CC labelled in yellow; two-tailed, unpaired t-test, not corrected for multiple hypothesis testing). **b** Representative immunoblotting shows cofilin hyperphosphorylation at serine 3 (and therefore reduced activity) (y axis denotes phospho-cofilin:total cofilin ratio; $N = 5$ control animals, 7 mutant animals; **$p = 0.0013$; two-tailed, unpaired t-test; error bars denote SEM). **c** Immunohistochemistry shows cofilin localisation to photoreceptor connecting cilium (CC) in wild-type retinas, partially lost in *Rpgr*$^{Ex3d8}$ mice, not seen in Cofilin knockout mice. White boxes define regions of interest depicted in bottom left panels (blue arrows highlight CC cofilin in wild type retina). Enlarged images (bottom right panels) highlight cofilin colocalization with CC actin in wild type photoreceptors. (Similar results seen in 2 independent experiments using separate animals). **d** Immunoprecipitation of wild type retinal lysates using magnetic beads coated (+) or uncoated (-) with cofilin antibody shows cofilin enrichment in bound fraction (bottom panel) and RPGR's retinal specific isoform, detected using GT335 antibody (black arrowhead, top panel; 52 kDa band is acetylated tubulin, bands at 50 kDa and 25 kDa are immunoglobulins. **e** Transmission electron micrograph of 8-week wild-type photoreceptors show OS composed of compacted discs extending to underlying retinal pigment epithelium (RPE) (left panel). Disc compaction is compromised in age-matched, *Cofilin* knockout photoreceptors, with discs appearing split and spaced out (right panel; similar results seen in 3 independent experiments). **f** Examination of basal discs shows an accumulation of shed vesicles (red arrow) in *Cofilin* knockout mice. (Scale bars; **c** = 10 μm top panels; 5 μm bottom left panels; 0.5 μm bottom right panels; **e** = 2 μm; **f** = 1 μm). Source data provided as Source Data file.

## Rpgr mutations lead to dysregulation of cofilin activity

Previous work by ourselves and others supports a role for RPGR in regulating photoreceptor actin[9,23,25]. To further explore this, we undertook an unbiased total proteomic approach in control and *Rpgr* mutant retinal extracts (Fig. 4a). At 3 months of age, computational analysis of differentially expressed proteins using Enrichr[31–33] ranked retinitis pigmentosa as the top disease in *Rpgr*$^{Ex3d8}$ retina compared to wild type control (Supplementary Fig. 7a). Further, analysis of protein pathways using Panther ranked cytoskeletal regulation by Rho GTPases as a significantly disrupted pathway (Supplementary Fig. 7a, Supplementary Data 1 and 2). Of note, there was altered expression of the ARP2/3 complex protein ARP3, known to play a role in disc morphogenesis[11]. To further explore this, immunoblotting was carried out on separate total retinal extracts, probing for a panel of actin nucleators and polymerisers. No changes were seen in levels of CDC42, profilin, VASP or WAVE proteins in *Rpgr*$^{Ex3d8}$ retinas (Supplementary Fig. 7b).

The ARP2/3 complex directly competes for binding sites on actin filaments with the actin depolymeriser cofilin, which we have previously shown to have reduced activity in *RPGR*-mutant patient-derived iPSC retinal organoid cultures[9] and is thought to be present in early discs[11]. To this end, we analysed *Rpgr*$^{Ex3d8}$ retinal lysates with immunoblotting, which showed increased phosphorylation, and therefore inactivation, of cofilin at serine 3 (Fig. 4b). Using immunofluorescence, we demonstrate that cofilin localises to the

photoreceptor CC (Fig. 4c), as was seen with RPGR localisation (Fig. 3f). Notably, this localisation was lost in *Rpgr*$^{Ex3d8}$ mice (Fig. 4c). We therefore sought to determine whether there is a biochemical interaction between RPGR and cofilin in vivo. Co-immunoprecipitation (co-IP) experiments using murine retinal lysates confirmed endogenous cofilin and the retinal specific isoform of RPGR occur in complex in the retina (Fig. 4d, Supplementary Fig. 7f)[34]. To clarify the relationship between RPGR and cofilin in photoreceptors, we generated a photoreceptor-specific *Cofilin* knockout mouse. Analysis of 8 week old mice using TEM showed the *Cofilin* knockout mouse phenocopies the 'spaced disc' phenotype seen in our *Rpgr*$^{Ex3d8}$ model. Their discs have lost compaction and appear spaced out (Fig. 4e). Further, we observed vesicle accumulation in the extracellular space at the base of outer segments (Fig. 4f), as seen in *Rpgr*$^{Ex3d8}$ mice.

Given actin's multifunctionality and its abundance throughout the cell, it is crucial that local, sub-cellular turnover is tightly controlled to ensure cellular processes are precisely executed, both temporally and spatially. Our findings support that RPGR occurs in complex with the actin-binding protein, cofilin, regulating its activity in the distal CC to control basal disc formation.

## Perturbation of RPGR dysregulates actin-mediated photoreceptor disc formation

Having demonstrated global dysregulation of actin-binding proteins in *Rpgr*$^{Ex3d8}$ mice, and compromised disc formation, we examined local

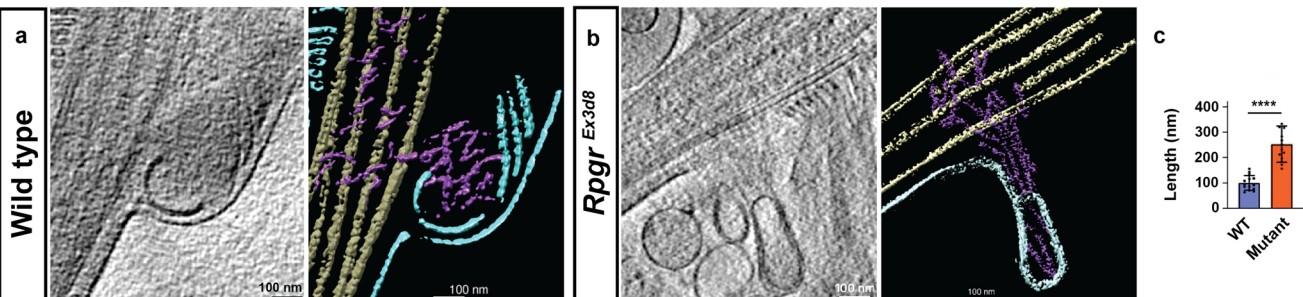

**Fig. 5 | Loss of *Rpgr* hyperstabilises actin in the basal photoreceptor disc. a, b** A projection from a tomogram (left panel) and a slice through a tomographic reconstruction (right panel) of the basal disc of a wild type (**a**) and *Rpgr*^Ex3d8 (**b**) photoreceptor to highlight the ciliary and disc membranes (cyan), microtubule-based axoneme (tan) and actin filaments (purple). **c** Actin filament length quantification in the basal disc of photoreceptor tomograms ($N = 4$ animals per group; ****$p \leq 0.0001$; two-tailed, unpaired t-test; error bars denote SEM). (Scale bars; **a**, **b** = 100 nm). Source data are provided as a Source Data file.

actin dynamics in the CC in the absence of RPGR. We confirmed increased F-actin was seen at the base of the OS in *Rpgr*^Ex3d8 compared to wild type (Supplementary Fig. 7c–e)[9]. To better resolve actin bundle integrity at the site of basal disc formation, *Rpgr*^Ex3d8 photoreceptors were further examined using cryoET. At 3 months of age, in keeping with reduced severing activity of cofilin, analysis revealed an increase in the length of actin microfilaments in *Rpgr*^Ex3d8 mice compared to wild type (Fig. 5a–c, Supplementary Movies 4 and 5). Binding of cofilin to F-actin is reported to cause shortening of the filament's helical pitch, thought to facilitate actin severing[35]. Whilst the shorter lengths of filaments in our raw tomograms limited our ability to draw definitive conclusions about individual filament pitch, actin in control rods was, on average, wider than *Rpgr*^Ex3d8 rods, suggesting differences exist in how actin-binding proteins decorate these filaments (Supplementary Fig. 8, Supplementary Movie 6).

In the absence of RPGR, we have shown increased levels and steady state lengths of F-actin in the region of the newest born discs, consistent with altered actin dynamics resulting in some stalled nascent discs being aborted and being shed as the observed vesicles/ectosomes (Fig. 2b). We hypothesised, therefore, that depolymerising actin in mutant retinas could rescue the disc biogenesis phenotype. *Rpgr*^Ex3d8 mice were treated for 6 h with the potent actin depolymeriser, cytochalasin D, by intravitreal injection of 0.5 µl at 25 mM. TEM analysis revealed a significant reduction in the number of vesicles shed at the base of the mutant outer segments with treatment (Fig. 6a, b), as well as an overgrowth of the newly formed basal discs, as previously described (Supplementary Fig. 9)[11,15]. Targeting photoreceptor ciliary actin dynamics with a broad-spectrum drug such as cytochalasin D, however, is not a therapeutic option, given the off-target effects that would likely result. We therefore sought to confirm if specifically overcoming the inhibition of cofilin activity, observed in *Rpgr* mutants, was sufficient to rescue.

LIM kinase 1 (LIMK1) directly phosphorylates at serine 3 and inactivates members of the cofilin family, resulting in stabilisation of F-actin. We treated wild type and *Rpgr*^Ex3d8 mice for 6 hrs with a LIMK inhibitor (TOCRIS CRT 0105950) by intravitreal injection of 0.5 µl at 100 µM. In treated wild type controls, TEM analysis revealed overgrowth of basal discs, phenocopying the effect of cytochalasin and confirming the pathway's key role in photoreceptor actin turnover (Fig. 6c). Furthermore, a significant reduction in the number of vesicles shed at the base of the mutant outer segments was observed in treated *Rpgr*^Ex3d8 mice (Fig. 6c, d). With these rescue experiments, we conclude that the shedding of aborted discs as vesicles, perturbing disc formation in *Rpgr* mutants, is due to compromised dynamic turnover of actin microfilaments extending from the photoreceptor CC into basal discs due to cofilin dysregulation. Our results suggest that this defect in disc biogenesis, resulting from increased length and stability of these

mutant actin microfilaments, can be therapeutically modulated in the short-term in vivo.

## Discussion

Our in vivo data support that RPGR functions locally at the photoreceptor CC to spatially regulate activity of actin-severing proteins, such as cofilin, in order to appropriately regulate actin-mediated disc formation. We have previously shown that RPGR interacts with another actin-severing protein, gelsolin, and these interactions are perturbed in pathogenic human *RPGR* retinal organoids[9]. We propose, therefore, that the glutamylated RPGR isoform serves as a structural scaffold that facilitates the timely recruitment and activation of actin-binding proteins (ABPs) to control disc biogenesis (Supplementary Fig. 10). Our study raises important questions about what the molecular nature of the pacemaker is that controls its regular tempo. Previous work at the primary cilia tip has demonstrated the highly localised role that actin plays in ciliary membrane deformation in a content-dependent manner. Here, excision of a membranous bud (ectosome) at its distal end, facilitates the exit of GPCRs to regulate cilia content appropriately[2,3]. Similar mechanisms at the distal CC/transition zone may operate to 'sense' regulate membrane content, such as rhodopsin concentration, to regulate disc formation. Further studies are necessary to distinguish these models. We propose that photoreceptor disc formation shares a common ancestral mechanism of ectocytosis and RPGR's retinal specific isoform may have evolved to promote this.

It is over 30 years since disc formation was first proposed to be an actin-dependent process. In seminal TEM experiments, it was shown that actin was present in the distal CC[14] and that depolymerising actin inhibited the formation of new discs, but not the addition of membrane to existing ones[15], suggesting actin bundling was required for the initial membrane deformation to begin the process. Subsequently, it was confirmed that the process of disc morphogenesis is one of evagination[36–38], which would support a process of active, actin protrusions initiating the process. In recent years, progress has been made in defining the molecular pathways that govern actin regulation in the CC[9–11,16]. Here, we confirm the presence of actin microfilament bundles within the basal disc.

Force generation lies at the heart of all cellular membrane morphogenesis, which the actin cytoskeleton can produce due to its ability to form higher order structures. But these resulting structures can remain stable for an incredibly long time[39]. In order for cells to perform the rapid actin turnover critical for a metronomic, repeated membrane deformation such as that required for disc formation, regulated dismantling of these networks is equally important. Thus, it would be predicted that a hyperstabilised actin cytoskeleton at the site of basal disc formation, as seen in our *Rpgr*-mutant mice, would compromise disc genesis. Interestingly, it was recently shown that disc renewal rates

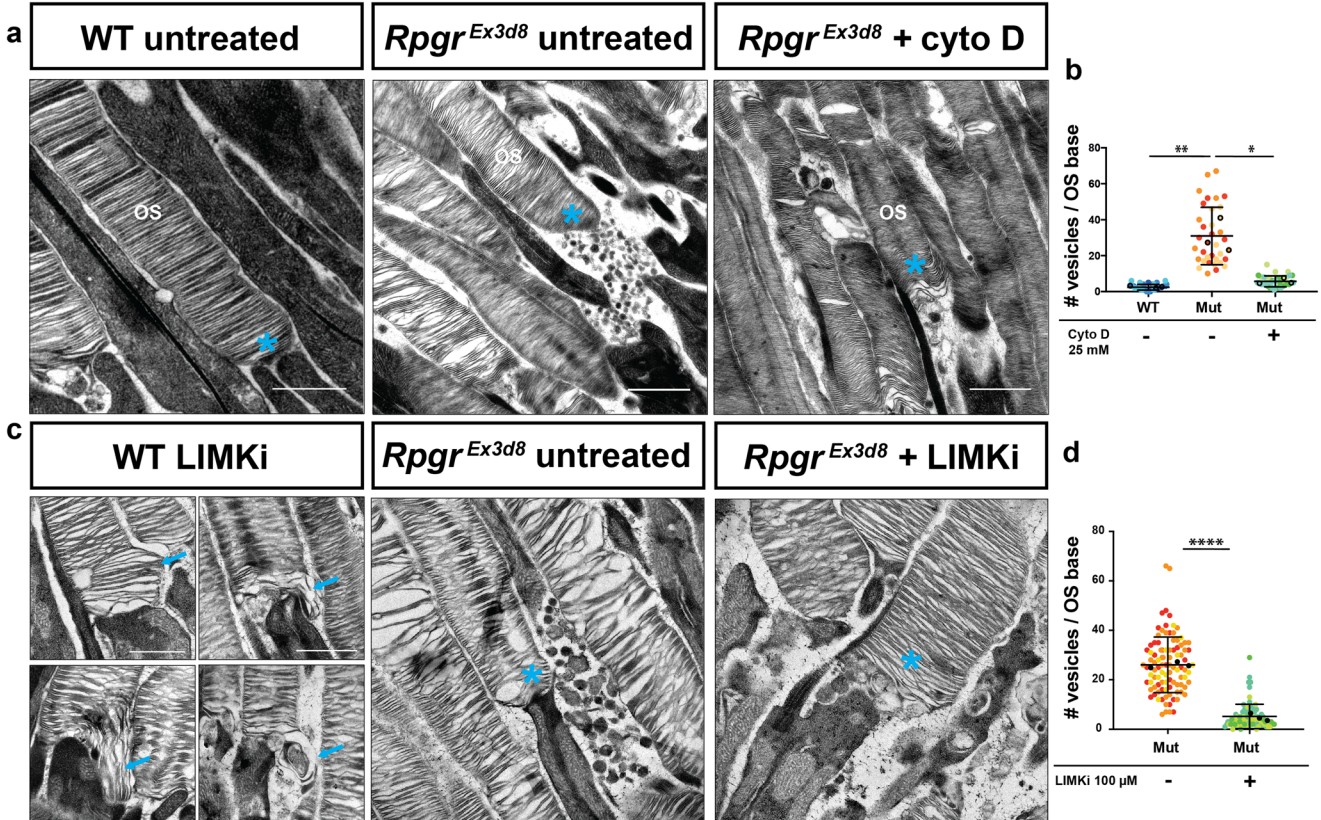

**Fig. 6 | Targeting actin severing pathways rescues shedding of vesicles from *Rpgr*-mutant photoreceptors. a** TEM reveals high numbers of vesicles shed from OS bases in *Rpgr$^{Ex3d8}$* photoreceptors (middle panel) compared to controls (left panel). Shedding of vesicles is reduced upon intravitreal delivery of 25 mM cytochalasin D for 6 h (right panel) (OS = Outer Segment; cyan * = OS base). **b** Quantification of vesicle shedding at the base of photoreceptors by TEM. (Black symbols = mean number of vesicles at the base of each photoreceptor per experimental animal; $N = 3$ animals per genotype; error bars represent standard error of the mean; **$p = 0.0068$, *$p = 0.0108$; two-tailed, unpaired t-test; error bars denote SEM. Different colours represent measurements from individual mice). **c** TEM reveals intravitreal delivery of 100 μM LIM kinase inhibitor for 6 h to wild-type retinas leads to elongation of basal discs (blue arrows, left panel). Intravitreal LIMKi delivery to *Rpgr$^{Ex3d8}$* eyes reduces number of OS shed vesicles (middle and right panels; cyan * = OS base). **d** Quantification of vesicle shedding by control or LIMKi treatment from TEM images. (Black symbols = mean number of vesicles at OS base of photoreceptors per experimental animal; $N = 3$ animals per genotype; error bars represent standard error of the mean; ****$p \leq 0.0001$; two-tailed, unpaired t-test; error bars denote SEM. Different colours represent measurements from individual *mice*). (Scale bars; a = 1 μm; c = 500 nm for 4 left panels of wild type treated photoreceptors; 1 μm for middle and right *Rpgr$^{Ex3d8}$* panels). Source data are provided as a Source Data file.

in the naturally occurring *Rd9* mouse, which harbours a 32 bp duplication mutation in the ORF15 of the RPGR retinal specific isoform (resulting in a premature stop codon), do not differ from wild type[40]. The two isoforms exist in complex[9] and more work is required to determine if they have discrete or overlapping functions at site of disc formation.

The actin-severing protein, cofilin, plays a central role in actin filament length oscillation by trimming the established end of a filament that continues to grow at the barbed end[39], allowing branched networks to be disassembled[41]. Of interest, cofilin competes for binding to actin with the actin polymeriser, ARP2/3, a protein previously demonstrated to be crucial for disc formation[11]. Cofilin is known to regulate primary cilia length through actin rearrangement[42]. Our work suggests cofilin is required for mammalian photoreceptor disc formation, and the observation that cofilin is present in aborted photoreceptor ectosomes when normal disc formation is disrupted[11] further supports this model. Further work is required to comprehensively define, spatiotemporaly, how actin-binding proteins localise to actin filaments as discs form.

The molecular architecture of this RPGR-dependent scaffold at the distal CC involves SPATA7 and RPGRIP[20], mutations in which also cause RP in humans. RPGR is mislocalised away from the distal CC in mice lacking either SPATA7[20] or RPGRIP[43], with failure of normal outer

segment disc genesis. Importantly, while RPGR, RPGRIP and SPATA7 appear critical for photoreceptor function, they appear dispensable for cilia function in most other cell types. We speculate, therefore, that they have evolved as a scaffold at the site of disc formation to which actin-binding proteins, critical for timely disc formation, can be recruited.

No treatment currently exists for any form of RP. Whilst gene therapy offers hope, long-term data for the only FDA- and EMA-approved gene therapy drug for genetic eye disease is less encouraging[44]. Complementary approaches must therefore be sought. Here, we show that pharmaco-manipulation of actin activity could reverse the underlying cellular phenotypes disrupting disc biogenesis. Further work is required to address the translational potential of our study, specifically to establish if long-term cofilin modulation could slow the retinal degeneration seen in *RPGR/XLRP* patients.

## Methods
### Animal experiments
All experiments followed international, national and institutional guidelines for the care and use of animals. Animal experiments were carried out under UK Home Office Project License PPL P1914806F in facilities at the University of Edinburgh (PEL 719 60/2605) and approved by the University of Edinburgh animal welfare and ethical

review body. Mice were housed at 19–24 °C with 45–65% humidity in a 12-h light-dark cycle (light on: 7:00 a.m.– 7:00 p.m). They received twice daily welfare checks by a trained and competent animal technician, with ad libitum access to food and water. Experiments were carried out blinded, where possible.

## Mouse generation

*Rpgr* mutant mice: Using CRISPR/Cas9 genome editing to target single cell embryos, deletion mutations were introduced into the *Rpgr* locus of C57Bl6/J mice (Supplementary Fig. 2). Ensembl *Rpgr* ENSMUSG00000031174, chromosome X X:10,175,515 was used for CRISPR design. The CRISPR target sequences (20-nucleotide sequence preceded by Bbs1 ends (Fd CACC / Rv AACA), followed by a protospacer adjacent motif [PAM] of 'NGG') were selected using the prediction software (www.crispr.mit.edu) and synthesised by Sigma, annealed and ligated into the pX330-U6-Chimeric_BB-CBh-hSpCas9 plasmid (Addgene 42230). The plasmid was transformed into chemically competent Stbl3 cells, colonies grown, maxi-prepped and correct sequencing confirmed by Sanger sequencing. The repair template oligo for *Rpgr*[ORFdS] (see below) was synthesised by Integrated DNA Technologies. See Supplementary Table 3 for list of guides and repair templates.

An 8 base pair deletion was introduced into exon 3 (c.296del8), resulting in a premature termination codon 68 base pairs downstream being shifted into frame and a predicted null protein (Supplementary Fig. 2); this mouse will hereby be referred to as *Rpgr*[Ex3d8]. In a separate round of injections, a 5 base pair deletion was introduced at the beginning of the repetitive domain of RPGR's retinal specific splice variant (c.2478del5), resulting in a frameshift and premature termination codon 52 base pairs downstream (Supplementary Fig. 2). This is predicted to result in a truncated protein lacking all of the protein's glutamate-rich consensus motifs (GEEEG), whose glutamylation is crucial for protein function[34]. This mouse will hereby be referred to as *Rpgr*[ORFdS].

Genome editing was performed by microinjection of the *Rpgr* guide / Cas9 plasmid (5 ng/µl) +/− a single-stranded repair oligo *Rpgr* (100 ng/µl) into single cells embryos. The injected zygotes were cultured overnight in KSOM for subsequent transfer to the oviduct of pseudopregnant recipient females[45]. From microinjection CRISPR targeting, genomic DNA from founder ear-clip tissue was subject to PCR screening and subsequent sequence verification of successful mutagenesis events by Sanger sequencing (Supplementary Fig. 2). Founder mice, mosaic for desired mutations were crossed with CD1 mice to generate large numbers of F1 progeny and allow unwanted allelic variants to segregate. Selected F1 mutants were backcrossed onto C57 mice and subsequently intercrossed. Initial genotyping was performed by PCR and Sanger sequencing using primers outlined in Supplementary Table 3. The genotyping of subsequent animals was performed by TransNetyx.

Western blot analysis showed loss of detection of a polyglutamylated protein running at the weight of *Rpgr*'s retinal isoform (Supplementary Fig. 2)[34]. In the absence of a commercially available antibody for murine RPGR, we consider this as confirmation of loss of protein expression.

Only male mice were used for experiments using *Rpgr* mice and their controls, due to the X-linked nature of the disease.

Generation of photoreceptor-specific *cofilin* knockout mouse: *Cofilin-1* floxed mice were a kind gift from Walter Witke (Bonn)[46], with LoxP sites previously inserted to flank the second exon of the *n-cofilin* gene of C57Bl6/J mice. Floxed deletion of exon 2 results in a frameshift and translational termination of n-cofilin. *Cofilin-1* flox mice were bred with the photoreceptor-specific Cre-recombinase mouse (iCre-75) on a C57Bl6/J background[47]. Mice homozygous for the floxed allele and having a Cre allele were used for experiments.

## Electrodiagnostic testing

All mice undergoing EDT were dark-adapted overnight prior to the procedure, and experiments were carried out in a darkened room under red light using an HMsERG system (Ocuscience). Mice were anesthetised using isofluorane and pupils were dilated through the topical application of 1% w/v tropicamide before being placed on a heated ERG plate. Three grounding electrodes were used subcutaneously (tail, and each cheek) and silver-embedded electrodes were placed upon the cornea held in place with a contact lens. The standard International Society for Clinical Electrophysiology of Vision (ISCEV) protocol was used, which recorded scotopic responses before a 10 min light adaptation phase in order to record photopic responses[48]. 3 and 10 cd.s/m$^2$ light intensity scotopic responses were used for analysis. Data was analysed using Graphpad Prism and compared by unpaired t-test.

## Optical Coherence Tomography (OCT) and blue light auto-fluorescence imaging

Mice were anesthetised using isofluorane and pupils were dilated through the topical application of 1% w/v tropicamide before being placed on a custom-built, heated plate on a Heidelberg Spectralis OCT machine. Mouse retina was brought into focus using IR imaging and fast volume scans (20 ° x 20 °) obtained (25 sections; 240 µm between sections) by spectral domain OCT using a 55 dioptre lens adapted for small animals. Accumulation of autofluorescent material in the retina, a clinical biomarker for retinal dysfunction, was assessed using the BluePeak™ blue light function on the Spectralis.

## Protein extraction, antibodies and immunoblotting

Retinas were lysed in 50 mM Tris pH8.0, 150 mM NaCl, 1 % NP-40 buffer containing protease inhibitors. Protein samples were separated by SDS-PAGE and electroblotted onto nitrocellulose membranes (Whatman) using the iblot System for 6 min (Invitrogen). Non-specific binding sites were blocked by incubation of the membrane with 5% non-fat milk in TBS containing 0.1% Tween 20 (TBST) for 1 h. Proteins were detected using the primary antibodies diluted in blocking solution (4% Bovine Serum Albumin in TBST) (Supplementary Tables 1 and 2 for list of antibodies). Following washing in PBST, blots were incubated with the appropriate secondary antibodies conjugated to horse-radish peroxidase (Pierce) and chemiluminescence detection of Super Signal West Pico detection reagent (Pierce) by high resolution image capture using the ImageQuant LAS4000 camera system (GE Healthcare). Images were transferred to Fiji/ImageJ and mean pixel intensity of protein bands measured for quantification, with an equal area of blot assessed across all bands.

## Rod outer segment proteomics

6 week old, dark-adapted mice were maintained in constant darkness for 12 h overnight prior to retina harvesting (also performed in the dark with infrared illumination). Rod outer segments were isolated by vortexing retinas in 150 µl of 27% sucrose solution in 20 mM HEPES, pH 7.4, 100 mM KCl, 2 mM MgCl$_2$ and 0.1 mM EDTA, spinning at 200 g for 30 s and collecting supernatant. Supernatants were diluted with the same buffer without sucrose and loaded in a step gradient of 27% and 32% sucrose prior to spinning at ~25,000 rpm and outer segment collection from the 32 to 27% sucrose interface. Proteins were solubilized in 2% SDS, 100 mM Tris·HCl (pH 8.0), reduced with 10 mM DTT (D0632; Sigma-Aldrich), alkylated with 25 mM iodoacetamide (I1149; Sigma-Aldrich), and subjected to tryptic hydrolysis using the HILIC beads SP3 protocol[11]. The resulting peptides were analyzed with a nanoAcquity UPLC system (Waters) coupled to an Orbitrap Q Exactive HF mass spectrometer (Thermo Fisher Scientific) employing the LC-MS/MS protocol in a data-independent acquisition mode. The peptides were separated on a 75 µm × 150 mm, 1.7 µm C18 BEH column

(Waters) using a 120 min gradient of 8 to 32% of acetonitrile in 0.1% formic acid at a flow rate of 0.3 mL/min at 45 °C. Eluting peptides were sprayed into the ion source of the Orbitrap Q Exactive HF at a voltage of 2.0 kV. Progenesis QI Proteomics software (Waters) was used to assign peptides to the features and generate searchable files, which were submitted to Mascot (version 2.5) for peptide identification. For peptide identification, we searched against the UniProt reviewed *mouse* database (September 2019 release) using carbamidomethyl at Cys as a fixed modification and Met oxidation as a variable modification. Proteins were included based on 3 criteria: 1) at least 1 peptide is identified in each independent experiment, 2) at least 2 peptides are identified in one experiment, and 3) the identification confidence score is > 95% in both experiments.

## Whole retina proteomics

6 week old, dark-adapted mice were maintained in constant darkness for 12 h overnight prior to retina harvesting (also performed in the dark with infrared illumination). Retinas were lysed in 2% SDS and kept at −80 °C until testing. 2 retinas from 1 *mouse* were used per biological replicate, with 5 biological replicates per group (mutant and wild type). In all cases, cell lysates were digested using sequential digestion of LysC (Wako) and trypsin (Pierce) using the FASP protocol (https://pubmed.ncbi.nlm.nih.gov/25063446/). Samples were acidified to 1% TFA final volume and clarified by spinning on a benchtop centrifuge (15 kg, 5 min). Sample clean-up to remove salts was performed using C18 stage-tips (Rappsilber et al., 2003). Samples were eluted in 25 μl of 80% acetonitrile containing 0.1% TFA and dried using a SpeedVac system at 30 °C and resuspended in 0.1% (v/v) TFA such that each sample contained 0.2 μg/ml. All samples were run on an Orbitrap FusionTM Lumos mass spectrometer coupled to an Ultimate 3000, RSL-Nano μHPLC (both Thermo Fisher). 5 μl of the samples were injected onto an Aurora column (25 cm, 75 μm ID Ionoptiks, Australia) and heated to 50 °C. Peptides were separated by a 150 min gradient from 5 to 40% acetonitrile in 0.5% acetic acid. Data was acquired as data-dependent acquisition with the following settings: MS resolution 240 k, cycle time 1 s, MS/MS HCD ion-trap rapid acquisition, injection time 28 ms. The data were analyzed using the MaxQuant 1.6 software suite (https://www.maxquant.org/) by searching against the *murine* Uniprot database (UP000000589_10090.fasta Downloaded 2018) with the pre-set settings (modifications, FDR and mass accuracy) enabling LFQ determination and matching between runs. The data were further analyzed using the Perseus software suite. LFQ values were normalized, 0-values were imputed using a normal distribution using the standard settings.

## Immunohistochemistry

Mice were sacrificed and eyes enucleated and placed into Davidson's fixative (28.5% ethanol, 2.2% neutral buffered formalin, 11% glacial acetic acid) for 1 h (cryosectioning) or overnight (wax embedding). For cryosectioning, eyes were removed from Davidson's fixative and placed into a series of consecutive 10%, 15% and 20% sucrose in PBS buffer for 15 mins, 15 min and overnight respectively for cryopreservation. Eyes were then embedded using optimal cutting temperature and kept at −80 °C until sectioned. Sections/cells were blocked/permeabilised with 4% BSA (Sigma) and 0.5% Triton X100 (Fisher) for 1 h at RT, washed with PBS, incubated with primary antibodies overnight at 4 °C, washed in PBS, incubated with secondary antibodies for 60 min at RT, washed in PBS, incubated in Hoechst for 5 min at RT and mounted with coverslips using Fluoromount-G (Southern Biotech). For wax preservation, eyes were fixed in Davidsons fixative overnight at 4 °C. Following fixation in Davidsons fixative, eyes were incubated successively in 70% v/v, 80% v/v, 90% v/v and 100% v/v ethanol, twice in xylene and then paraffin, each for

45 min per stage, using a vacuum infiltration processor. Hematoxylin and eosin staining was performed on 8 μm paraffin tissue sections and imaged on a Zeiss Brightfield microscope. Confocal imaging was performed on a Nikon Eclipse TiE inverted microscope with Perfect Focus System using resonant scanning mirrors (equipped with 405 nm diode, 457/488/514 nm Multiline Argon, 561 nm DPSS and 638 nm diode lasers) with detection via four Photomultiplier tubes (2x standard Photomultiplier tubes and 2x GaAsP PMTs). A 60x oil immersion lens was used. Data was acquired using NIS Elements AR software (Nikon Instruments Europe, Netherlands). Z-stacks were processed and analysis in Fiji/ImageJ.

## Airyscan imaging

Sample preparation for Airyscan confocal immunofluorescence microscopy. Mice 6 weeks of age were transcardially perfused with 80 mM PIPES, pH 6.8, 5 mM EGTA, 2 mM MgCl₂, 4% paraformaldehyde). Eyes were enucleated, and after the cornea was removed, they were immersion fixed overnight at 4 °C. After removal of lens, the eyecups were flash-frozen in optimal cutting temperature using liquid nitrogen. 8 μm cryosections were collected and stained for rhodamine wheat germ agglutinin (Vector RL-1022) and phalloidin conjugated to Atto647N (Sigma 65906) or cofilin (CST 5175) and phalloidin conjugated to Atto647N (Sigma 65906). The sections were then mounted in Prolong Glass (Invitrogen P36980). Sections were imaged on a Zeiss LSM 880 Airyscan Fast Confocal Microscope using a 63x objective. Z-stacks were first processed in Zeiss ZenBlue software for Airyscan processing, then colour processing and analysis were performed in Fiji/ImageJ. Actin puncta quantification was performed by taking a 0.35 μm ROI around each actin puncta that was at the base of an outer segment, slice by slice, on each z-stack. Then the averaged relative integrated density of three background ROI's were subtracted from each relative integrated density measurements from the actin ROIs. These measurements were then plotted in Prism software, where Students t-test statistical analysis was performed.

## Transmission electron microscopy

6 week old mice were euthanized by transcardial perfusion using fixative (50 mM MOPS, pH 7.4, 2% glutaraldehyde, 2% paraformaldehyde, 2.2 mM CaCl₂). Eyecups were enucleated and placed in 1 mL of fixative. After 30 min, the cornea and lens were removed, then left to incubate in fixative for a total of 2 h at room temperature. Thereafter, retinas were either:

1. Washed in 0.1 M phosphate buffer (pH 7.4), postfixed with 1% osmium tetroxide (Electron Microscopy Science) and dehydrated in an ethanol series prior to embedding in Medium Epoxy Resin (TAAB). Ultrathin (75 nm) sections of the retina were then stained with aqueous uranyl-acetate and lead citrate and then examined with a Hitachi 7000 electron microscope (Electron Microscope research services, Newcastle University Medical School).

2. Polymerized in 4% agarose (Genemate E-3126-25). 150 μm sections were collected into MilliQ water using a vibratome. Sections were then stained in 1% tannic acid in 0.1 M HEPES, pH 7.4, for 1 h, with rocking at room temperature, covered. After rinsing in MilliQ water, the sections were stained with 1% uranyl acetate in 0.2 M maleate buffer, pH 6.0, for 1 h, with rocking at room temperature, covered. The sections were rinsed in MilliQ water and dehydrated in a series of ethanol washes (50%, 70%, 90%, 100%, 100%) for 15 min each, followed by two 100% acetone washes, 15 min each. The sections were then embedded in Epon-12 resin by sandwiching the sections between two sheets of ACLAR (EMS 50425-10) and leaving them at 60 °C for 48 h. Ultrathin silver sections (~ 60 nm) were placed on copper slot grids (EMS FF2010-CU) and

poststained in 1.2% uranyl acetate in MilliQ water for 6 min, followed by staining in Sato's lead (a solution of 1% lead acetate, 1% lead nitrate, and 1% lead citrate; all from Electron Microscopy Sciences) for 2 min. Sections were imaged on a JEOL JEM-1400 electron microscope.

### Cytochalasin rescue experiment

Mice were anesthetised using isofluorane and pupils were dilated through the topical application of 1% w/v tropicamide. 0.5 μl of 25 mM Cytochalasin D (Sigma) in PBS was injected intravitreally. Contralateral eyes had a sham injection of 0.5 μl PBS and served as the control eye. 6 h later mice underwent transcardial perfusion using fixative and were processed for electron microscopy as outlined above.

### LIMKi rescue experiment

Mice were anesthetised using isofluorane and pupils were dilated through the topical application of 1% w/v tropicamide. 0.5 μl of 100 μM LIMK inhibitor (TOCRIS CRT 0105950) in PBS was injected intravitreally. Contralateral eyes had a sham injection of 0.5 μl PBS and served as the control eye. 6 h later mice underwent transcardial perfusion using fixative and processed for electron microscopy as outlined above.

### STORM immunohistochemistry and resin embedding

Unfixed retinas from 6- to 12-week old wild type mice were immuno-labeled for STORM using a protocol previously developed[49]. Most 1 μm resin sections were collected into 35 mm glass-bottom dishes, and 2 sections for each sample were also collected onto a #1.5 coverslip, mounted in Prolong Glass (Invitrogen P36980) and imaged on a Structured Illumination Microscope (SIM); DeltaVision OMX Blaze v4 (GE Healthcare, now Cytiva); a PLANPON6 60×/NA 1.42 (Olympus) using oil with a refractive index of 1.520. Z-spacing of 125 nm was used for each 1 μm z-stack. SIM reconstructions and alignment were performed in Softworx 7 software. After analysis, reconstructions were processed in Fiji/ImageJ.

Immediately prior to STORM imaging, 10% sodium hydroxide (w/v) was mixed with pure 200-proof ethanol for 45 min to prepare a mild sodium ethoxide solution. Glass-bottom dishes with ultrathin retina sections were immersed for 30–45 min for chemical etching of the resin. Etched sections were then washed and dried on a 50 °C heat block. The following STORM imaging buffer was prepared: 45 mM Tris (pH 8.0), 9 mM NaCl, and oxygen scavenging system: 10 mM Sodium Sulfite, 10% (w/v) dextrose + 100 mM MEA (i.e., L-cysteamine, Chem-Impex) + 10% VECTASHIELD (Vector Laboratories). Imaging buffer was added onto the dried, etched sections and sealed with a second #1.5 coverslip for imaging. Imaging was performed on the Nikon N-STORM system, which features a CFI Apo TIRF 100× oil objective (NA1.49) on an inverted Nikon Ti Eclipse microscope. Photobleaching and photo-switching initiation were performed using both the 561 and 647 mm laser lines at maximum power. Imaging frames were collected at 30 frames per second. A total of 40,000 frames were collected for each imaging experiment.

### STORM image analysis

Two-dimensional (2D) STORM analysis of STORM acquisition frames was performed using NIS-Elements Ar Analysis software. Analysis identification settings were used for detection of the individual point spread function (PSF) of photoswitching events in frames from both channels to be accepted and reconstructed as 2D Gaussian data points. These settings were as follows: minimum PSF height: 400, maximum PSF height: 65,636, minimum PSF width: 200 nm, maximum PSF width: 700 nm, initial fit width: 350 nm, maximum axial ratio: 2.5, maximum displacement: 1 pixel. After analysis, reconstructions of single cilia were processed in Fiji/ImageJ for straightening.

### Isolation of Rod Outer Segments (ROS)

Mice were maintained in a 12/12 h light (400 Lux)/dark cycle. Wild-type C57BL/6 J, were purchased from Jackson lab (Bar Harbor, ME). 3-month males of $Rpgr^{Ex3d8}$ and C57BL/6 J mice used for tissue samples were euthanized by $CO_2$ inhalation prior to dissection following American Association for Laboratory Animal Science protocols. Preparation of purified *mouse* ROS was modified from[50,51]. Briefly, freshly dissected retinas were collected in 200 μl Ringer's buffer (10 mM HEPES, 130 mM NaCl, 3.6 mM KCl, 1.2 mM $MgCl_2$, 1.2 mM $CaCl_2$, 0.02 mM EDTA, pH 7.4) with 8% (v/v) OptiPrep (Sigma) under dim red light. Retinas were pipetted up and down with a 200 μl wide orifice tip 50 times, and then centrifuged at $400 \times g$ for 2 min at room temperature. The supernatants containing ROS were collected on ice. The process was repeated 4–5 times. All samples were pooled and loaded onto the top of 10, 15, 20, 25 and 30% (v/v) OptiPrep step-gradient and centrifuged for 60 min at $19,210 \times g$ at 4 °C in a TLS-55 rotor (Beckman Coulter). The ROS band was collected with a 18 G needle, diluted with Ringer's buffer to 3 ml, and pelleted in a TLS-55 rotor for 30 min at $59,825 \times g$ at 4 °C. The ROS pellet was resuspended in Ringer's buffer, and this suspension was applied to grids for cryo-electron microscopy.

### Cryo-electron tomography and image processing

Rod cell fragments containing outer segments, CC, and portions of the IS were collected from WT and $Rpgr^{Ex3d8}$ mice by iso-osmotic density-gradient centrifugation and applied to EM grids as described above and previously[19,49,51]. Briefly, isolated WT or $Rpgr^{Ex3d8}$ cilia were mixed with BSA-stabilized 15 nm fiducial gold (2:1), and 2.5-3 μL of the mixture was deposited on the freshly glow-discharged Quantifoil carbon-coated holey grids (200 mesh, Au R3.5/1) and blotted from the front-side or backside before plunging in liquid ethane using a Vitrobot Mark IV (FEI, Inc.) or automated plunge-freezing device Lecia EMGP (Lecia, Inc.). The frozen-hydrated WT and $Rpgr^{Ex3d8}$ specimens were stored in liquid nitrogen before imaging. The frozen-hydrated samples were imaged on a JEOL 3200FSC electron microscope operated at 300 kV using a K2 Summit direct electron detector camera or on a JEOL JEM2200FS operated at 200 KV microscope with a DE12 direct electron detector. Both microscopes were equipped with a field emission gun, an in-column energy filter. Single-tilt image series were automatically collected using SerialEM[52] at a defocus range of 8–12 μm and a magnification of 12,000x (equivalent to 3.6 Å/pixel, WT) and 15,000x (equivalent to 4.2 Å/pixel, KO) on each microscope. The total electron dose per tomogram was 80–100 electrons/Å², as typical for cellular cryo-ET. Each tilt series, for WT, has 35 tilt images covering an angular range −51° to +51° with 3° increment ( ± 51°, 3° increment); for $Rpgr^{Ex3d8}$, there were 51 tilt images with 2° increment ( ± 50°, 2° increment). Tilted images were automatically aligned and reconstructed using the automated workflow in EMAN2 software[53]. Tomographic reconstructions and 3D surface rendering of sub-tomogram averages were generated and visualized using IMOD[54] and UCSF Chimera[55] (http://www.rbvi.ucsf.edu/chimera).

For the segmentation, to enhance the contrast, tomograms were averaged by two (bin2) or four (bin4) voxels and filtered uniformly using a low-pass filter (set up at 80 Å) to reduce the noise. Structural features such as actin filaments, MT and PM were manually annotated using IMOD[56,57] and *Scripts* (e2maskimod.py) from EMAN2[58] software package. The segmented maps were visualized using UCSF Chimera 3D software package (http://www.rbvi.ucsf.edu/chimera).

### Subtomogram averaging of actin filaments

The actin filaments were picked from 4x binned tomograms before finer alignment with 1x binned data. Approximately 700 volumes of $60 \times 60 \times 60$ nm actin filaments were extracted at the base of ROS region from WT and $Rpgr^{Ex3d8}$ tomograms. The initial models (low-pass

filtered to 60 Å) were generated using EMAN2 routine[59–61] without applying any symmetry. The subsequent iterative subtomogram refinement using EMAN2[53] yielded the averaged structures of actin filaments at ~30 Å resolution. The refinement was performed in 'gold-standard' fashion with all particles randomly split into two subsets with resolution measured by the Fourier shell correlation of the density, using the 'gold' standard cut-off criterion of FSC = 0.143. For the WT samples, approximately 400 volumes of 60 × 60 x 60 nm actin filaments were extracted at the base of ROS region from five tomograms. For $Rpgr^{Ex3d8}$, approximately 350 volumes of 60 × 60 x 60 nm actin filaments were extracted at the base of ROS region from 6 tomograms. The previously reported F-actin periodicity of 5.5 to 5.9 nm was observed in initial models for both WT and $Rpgr^{Ex3d8}$. The results obtained with these initial models were consistent with the results obtained when using previous F-actin models (with ~37 nm, and ~27 nm half helical repeat lengths and low-pass filtered to 60 Å)[62] as initial references to initiate iterative refinement of subtomogram-averaged models.

### Quantification and Statistical analysis
All statistical analysis was carried out using GraphPad Prism 8 (version 8.4.1; GraphPad software, USA) as described in the text. To determine statistical significance, unpaired t-tests were used to compare between two groups, unless otherwise indicated. The mean ± the standard error of the mean (SEM) is reported in the corresponding figures as indicated. Statistical significance was set at $P \leq 0.05$.

### Reporting summary
Further information on research design is available in the Nature Portfolio Reporting Summary linked to this article.

## Data availability
All data are available in the main text or the supplementary materials. Proteomics data have been deposited to the ProteomeXchange Consortium via PRIDE and are available via ProteomeXchange with identifier PXD038015. Materials and reagents are available from the corresponding author upon request. Source data are provided with this paper.

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

## Acknowledgements

We are grateful to the IGC Advanced Imaging Resource, the IGC Mass Spectrometry facility and HGU Bioinformatic Analysis Core, as well as the University of Edinburgh and Newcastle University Electron Microscopy Research services. We acknowledge Hemant Khanna for providing his RPGR antibody. We acknowledge the proteomic facility at the Duke Eye Centre for carrying out the 6 week old rod outer segments mass spec experiment. We thank Vadim Arshavsky for sharing his PRCD antibody and protocols, access to core facilities and for specialized mass spec analysis. This work was funded by the Wellcome Trust (RM; 219607/Z/19/Z, FM; 215343/Z/19/Z, MJ; ISSF, AvK; 208402/Z/17/Z), the National Institute for Health (AM; F32 EY031574, TGW and ZZ; R01-EY026545, TGW and FH; R01-EY031949), Fight for Sight (FN; 5179 / 5180), the Medical Research Council (PM and LM: MC_UU_00007_14, MR_Y015002_1, MC_PC_21044), Cancer Research UK (LMM; A24452) and the Welch Foundation (TGW; Q-0035).

## Author contributions

R.M. and P.M. conceived the project. R.M., A.M., Z.Z., L.C.M., L.M.M., T.G.W. and P.M. designed the experiments. R.M., A.M., Z.Z., F.N., F.M., L.C.M., Av.K., L.M., F.H., M.K.J., P.A.T. and C.B. performed the experiments. R.M., A.M., Z.Z., F.N., F.M., L.C.M., Av.K., T.G.W. and P.M. analysed the data. R.M., L.M.M., T.G.W. and P.M. coordinated the study and provided guidance. CG and AKG shared reagents or supporting models used in this manuscript. R.M., A.M., L.M.M., T.G.W. and P.M. wrote the paper. All of the authors discussed the results and approved the final version of the manuscript.

## Competing interests

The authors declare no competing interests.
