## [Peer Review File · Nature Communications]

Ciliary tip actin dynamics regulate photoreceptor outer segment integrityREVIEWER COMMENTS

Reviewer #1 (Remarks to the Author):

The manuscript shows an elegant series of experiments on a relevant topic with significance to the research field. Several very convincing data are presented. Though a few of the conclusions seem premature and additional evidence is needed.

- Fig 3e legend should mention how many independent experiments (nr of mice) intravitreally injected, how many sections analysed, inform what the error bars represent (SD or SE), and what the 3 stars indicate as p-value, and indicate what the different colours of blue and orange represent. The pictures Fig3c and 3d show different thickness of ONL. Does the intravitreal delivery of the blocking agent influence the function of rhodopsin and therefore vision in the mice? Furthermore, there is missing an essential control in these experiments. Authors should proof that the lengths of outer segments are not changed during the block-chase experiments. Data should be shown after 3-days chase without labelling after 3 days versus 3-days chase with labelling after 3 days. The latter should give no significant differences whereas the first mentioned should. If no differences, the authors should discuss the ongoing phagocytosis by the RPE. Therefore, the conclusion that mutations in Rpggr reduce the rate of photoreceptor disc formation are premature.
- Claimed overlap of cofilin with RPGR is not demonstrated by showing RPGR staining in Fig3a and cofilin staining in Fig4d.
- Fig 4e legend should mention the statistical data. How often have these experiments been performed independently? It seems that there is nonspecific binding of GT335 interactive material, suggesting that the band at ~200kDa (?) is a-specific. In other words, there is no proof for direct binding of cofilin to glutamylated-RPGR. Furthermore, the most essential control is lacking in these experiments (retinal lysates that lack the RPGR protein such as retinal lysates from RpggrEx3d8).
- Fig 6 legend should better explain the statistical analysis of these studies depicted in fig6d.

Minor

- The text in the manuscript and the legend to Fig4 do not mention the 4 labelled dysregulated actin binding proteins (two proteins lowered, two proteins increased in levels in the mutant). The mass spec data needs to be deposited in a bank and should become accessible at publication.
- The protocol to detect the 6kDa PCD protein should be described or referred to.
- The manuscript should more clearly describe that there should be no RPGR protein in the RpggrEx3d8 strain. Fig3a suggests that there should be no RPGR but there is still a lot of signal which according to the authors is due to aspecific centrosomal staining, without further proof.
- In figure 4d, the authors claim that cofilin localises to the connecting cilium, but do not use a ciliary marker to confirm this. The authors should show the proof to this or provide the reference where this colocalization has been shown previously. Also the authors do not show the localisation and presence of cofilin in mutant retina. Is cofilin correctly localised to the CC and at normal levels?
- The authors do use the presence of rhodopsin as a measure for raw number of outer segment discs. The

authors should provide proof for the correctness of such statement because young mice lacking rhodopsin function temporarily have outer segments with discs. Otherwise, the authors should reference to the literature that photoreceptor discs without rhodopsin do not exist. Or at least discuss that such discs do exist but are less stable.

-The video (433804_1_supp_7764101_rwndn3) provided that shows SNAP-TMR, SiR-Actin and Hoechst staining, a comparison between wild type and RpgrEx3d8 is potentially not representative. The wild type does not show a normal layer of ONL, therefore not showing a representative localization of F-actin in the wild type (and overrepresenting the levels of F-actin in the mutant retina).

-Did any of the block-chase experiments result in relocalisation of rhodopsin-SNAP in the ONL?

-Are ectosome vesicles observed as well in RPGR mutant human retina or would the findings be mouse specific?

- In the proteomics studies there was altered expression of ARP3. Was this tested on Western blots? The authors report a small change in expression of ARP2.

- Data Fig5d performed blind for genotype ? By at least two independent researchers ? - Data Fig5e seems (overly) convincing but could as well be in part due to inappropriate retinal slice culturing. Blind for genotype, at least 2 researchers. What time of culturing and measurement?

- Extended data Fig 3 does not show loss of photoreceptor function. It only shows morphological studies. The extended data Fig 4 does show the loss of retinal function. Change the title of extended data Fig 3.

- Extended data Fig 6 suggests that the biosensor rhodopsin-SNAP cause decreased outer segment length upon SAP-block

- Ext data Fig 6 is unclear. Were the retinal slice in ext data Fig6f naïve retinal slice culture (without SNAP-block) as well cultured for 72 h whereafter stained with fluorescent SNAP ligand? Needs better presentation what the experiment was:

6f (no-block): No intravitreal SNAP-block & SNAP ligand after 72 h culturing

6f (block-chase): Intravitreal SNAP-block & SNAP ligand after 72 h culturing.

Not sufficient clear that this is consistent with newly formed discs. It might just interfere with rhodopsin function. In this experiment with wild type mice it has nothing to do with RPGR?

- Seems as if f-actin is present in the ectosomes?

- Knockout of RPGR still actin binding protein activity >> slows down how much?

Reviewer #2 (Remarks to the Author):

The paper by Megaw et al. entitled "Ciliary tip actin dynamics regulate the cadence of photoreceptor disc formation" offers much exciting news on the dynamics of disc morphogenesis via the control of actin cytoskeletal activity. It is well written and well-illustrated using several state of the art techniques. Although the involvement of actin in rod disc development has been known for a long time the mechanism is largely unknown. The role of Rho-related proteins in the mechanism is novel and timely. Retinal disease continues to be in need of effective therapies and this can only be accomplished by continued progress in understanding the normal mechanism and how mutant genes can subvert normal function. This paper adds to both information about normal and pathological function.

I have only minor suggestions.

Ln 196 Fig 3a was confusing. Indications, in the images, of the location of CC and centrosome would be helpful. Also the non-specific staining using RPGR antibody seems different in the wild type and the mutant. Perhaps an extended figure showing the non-specificity would be helpful to show that the difference is purely a technical issue.

Ln 292 Please identify if “n” is animals or something else.

Ln 315 Please include procedure for LIMKi use in the methods. Also it would be helpful for comparison with other past or future work and for possible translational studies to estimate what the concentration of LIMKi and cytochalasin is in the mouse eye.

Ln 482 & 568 The abbreviation “OCT” is used for two different things- optical coherence tomography and optimal cutting temperature. Please change the abbreviation for one of these.

Ln 550 Change “to” to “for”

Ln 554-557 Please rewrite this sentence. Does not make sense.

Ln 598 I think “Grids” should say “sections” for more accuracy.

Ln 603 Please add “were” processed

Ln 649-650 Please remove italics

Ln 690 Remove the word “frozen”

Ln 706 A space might be added between (bin4) and voxels

In general, the methods section should be carefully reviewed as this reviewer only looked at methods that were known to them.

Reviewer #3 (Remarks to the Author):

In this manuscript the authors have explored the role of RPGR, a protein encoded by the RPGR gene linked to X-linked retinitis pigmentosa, in photoreceptor disc morphogenesis. Using a variety of cell biology techniques including CryoEM tomography, live imaging, transgenic mice, and immunocytochemistry and immunoprecipitated, they have confirmed the presence of actin containing microfilaments within the newly forming discs at the base of the outer segment and showed that this process of disc maturation is slowed in mice containing either a frameshift or truncation mutation in the retina-specific isoform similar to that observed in humans with XLRP. Analysis of these mice together with immunolocalization and immunoprecipitated has led the investigators to conclude that RPGR regulates actin dynamics through binding the actin severing protein cofilin in the cilia and mutations in RPGR compromise the timing of disc morphogenesis related to actin dynamics.

Much of this study has confirmed earlier studies showing that actin plays a key role in disc morphogenesis at the base of the outer segment. CryoEM has shown the presence of actin containing microfilaments within the newly forming disc evaginations and the authors provide evidence that the process of outer segment formation is hindered in RPGR mutant mice (shortening of the outer segment and production of exomes) with mutations in ORF15 – thereby providing confirmation for the role of RPGR in disc morphogenesis. Indeed, the authors have previously reported that RPGR activates the actin-severing protein gelsolin, a function associated with actin disassembly in the cilium and disease-

causing RPGR mutations perturb this RPGR-gelsolin interaction.

In a key aspect of this manuscript, the authors suggest that RPGR interacts with cofilin based on co-immunoprecipitation and co-localization results. However, the data is not overly convincing as it is based on a single coimmunoprecipitation result shown in Figure 4e. The blot itself is of relatively poor quality and co-immunoprecipitation does not necessarily prove that there is a direct interaction of RPGR and cofilin. If one were to believe this co-immunoprecipitation experiment, then it is possible that the cofilin antibody is pulling down a large complex containing RPGR, but not necessarily binding directly to RPGR. More direct interaction and the effect of RPGR in cofilin activity is needed to draw a conclusion regarding the interaction of cofilin and RPGR and the effect of RPGR on cofilin actin depolymerizing activity. The manuscript would be strengthened if the authors could isolate RPGR and cofilin and show that they directly interact and that RPGR modulates the depolymerization activity of cofilin.

Other issues:

Figure 2: It would help the reader if a higher magnification image were presented to show the differences in disc compaction and spacing by TEM.

It is unclear why the authors use PRCD has an indicator to show that rod outer segments of Rprg(Ex3d8) retinas contain lower photoreceptor disc component than WT retina. Proteomic data should show a decrease in all the outer segment specific membrane proteins including rhodopsin, peripherin2, ABCA4, etc. This should be added to the manuscript to provide the reader with confirmation on the difference in length of the outer segments.

Figure 4e: There are many bands in the immunoblot of RPGR. The authors should inform the reader what these bands represent. If they are crossreacting proteins, then this could comprise the studies involving immunofluorescence microscopy and would suggest that the antibody are not overly specific. As indicated above, it would be important to show a direct interaction or regulation of cofilin by RPGR such as using heterologous expression system together with co-IP and activity assays.

In the discussion, it is unclear how the interaction of RPGR with gelsolin relates to the potential interaction of RPGR with cofilin. It would be helpful if the authors discuss how RPGR may regulate actin dynamics via these actin severing proteins.

Reviewer #4 (Remarks to the Author):

The formation and renewal of photoreceptors outer segment discs is essential for vision. The processes underlying the generation of discs is not well understood making this an important area of research. Megaw, et al have used a range of imaging approaches as well as taking advantage of disease and reporter mouse models to investigate this process. In this paper they propose actin dynamics regulate the processes that underlie new disc formation and elongation. Evidence for the role of actin in disc formation has been shown in previous studies, including recent work from the Arshavsky lab, but has not been visualised at a resolution that can be achieved by cryo electron tomography. Furthermore, in this work the authors propose RPGR controls disc formation by binding to cofilin.

My main concern of this study is the preservation of the samples used for electron microscopy and tomography. Many of these samples appear suboptimal or it is difficult for the reader to determine the level of preservation based on how the data is presented. This is particularly relevant for the tomography data as a single slice from each tomogram is shown of a small area of the photoreceptor. Supplementary videos of the tomography data should be included so that the readers can see each slice from the full data stack. I also have a number of other concerns about the data within the manuscript that I have listed below.

It would help to have lower magnification cryo electron microscopy images in the main manuscript or supplementary data to show more photoreceptors and provide evidence that they are preserved well enough to be assessed by cryo electron tomography. The reviewer acknowledges that preserving photoreceptors for cryo-electron microscopy is difficult and there are limitations, but it is important to be sure that the photoreceptors are not damaged, and the membrane have not been distorted and actin filaments displaced.

In Figure 1 the tomography data shows an odd-looking structure (due to its size, shape and contents) described as basal discs. From the single slice in (b) there appears to be membranes that are not included in the segmentation. Due to this and potentially other structures being excluded, the segmentation does not seem to match up well with the tomography slice. Either more of the membrane should be segmented to reflect the tomography data or there needs to be more clarity about what has been included or excluded from the segmentation in (b).

In Figure 1 panels (e) and (f), it is difficult to compare the subtomographic reconstruction against the previously published actin structure. It would be helpful to have movies to show these filaments rotating around the y-axis.

Extended Data Movie 1 shows some slight actin movement, but this could be the actin within the connecting cilium, rather than the basal discs. The photoreceptor also appears to be moving too, so it is hard to draw conclusions about the actin dynamics from this data.

Based on the results shown in Figure 1 there is not enough evidence to say 'We thus conclude that dynamic actin changes occur within nascent photoreceptor discs, suggesting a role for actin in disc genesis'. As it is not clearer that the structure containing actin are basal discs and actin polymerisation can not be clearer assessed from the time lapse videos.

In Figure 2 (a) the wild-type panel is not a good representation of the OS length as photoreceptor are slightly oblique and entire OS are not shown. Any obliqueness of the OS would impact on the measurements made in (b). It would also be better to show images that include the edge of the RPE, to clearly show the full length of the OS and that they are shorter in the RprgrEX3d8 model.

The authors should be careful about using the term 'split disc' phenotype, as this can be a sample preparation artefact rather than a genuine phenotype. An example of this is in Figure 6 (a) for the WT untreated sample there are greater gaps between some of the discs that are not as evident in the better-

preserved WT sample in Figure 2 (c).

When examining the images in Figure 2 (a) and (b) the OS could potentially be wider in RpgrEX3d8 model. It would be good to check to this to be certain that only the length is affected.

In the text or in the legend of Figure 3, there needs to be more information about the polyglutamylation staining to say that is used to label polyglutamylated tubulin. Furthermore, is the polyglutamylation staining only in the connecting cilium or does it extend into the OS? From the images in Figure 3a it seems that RPGR does not extending as far as the polyglutamylation staining and therefore, may not extend all the way up the connecting cilium to the site of disc formation.

If actin filaments extend into newly forming discs are these released in ectosome in the RpgrEX3d8 model? Can you see filaments or other structures in the ectosomes when examined by cryo-electron tomography (Figure 5 (b)), or by room temperature electron microscopy or tomography?

When looking at Extended Figure 7 the OS of the RpgrEX3d8 model appear to be longer than the WT. The WT is perhaps slightly oblique but according to Figure 2b the OS should be approximately 25% shortening of the OS in the RpgrEX3d8 model, which not evident in this data.

In Figure 4 the measurements from the Western blots in (b) and (c) are not convincing. In (b) the α -tubulin looks over exposed and there is no visible difference between the bands for ARP2 and phosphor Cofilin. This is unlike Figure 2 (e) and (f) where you have a clear visible difference in the PRCD bands.

As a comparison to the Cofilin staining in Figure 4 (d) it would be good to show the staining in the RpgrEX3d8 model. Co-staining of RPGR and Cofilin should be included to show colocalization between the two. It would also add to the paper to show co-staining of actin and Cofilin to see if there is complete colocalization of the two within the connecting cilium and base of the OS.

As mentioned above for Figure 5 it is important to include supplementary videos of the tomograms and provide more images to show that the samples have been adequately preserved and to allow the reader to see the length of the actin filaments. In extended movie 2 the preservation of the wild type is very poor compared to RpgrEX3d8. If this was used for the analysis in figure 5 (d), the preservation is too poor to be used. Furthermore, there seems to be very little actin movement in the movie, with most actin movement occurring within inner segment region.

In the text there is mention that there is disc overgrowth when treating mice with cytochalasin D, but this is not evident in Figure 6. An image of WT mouse treated with cytochalasin D should be included so that it can be compared to the RpgrEX3d8 model.

In Extended Data Figure 8, the correlation coefficient calculated when comparing the wild type and RpgrEX3d8 subtomographic averages is not very meaningful, considering part of the wild type reconstruction is poorly resolved and most of the overlapping structure is the central region of the filaments. The wild type structures with and without imposed helical symmetry looks different from the structures shown in Figure 1. Where these generate from a different set of tomograms?

Reviewer #1:

(#1) Fig 3e - mention how many independent experiments (nr of mice) intravitreally injected, how many sections analysed, inform what the error bars represent (SD or SE), and what the 3 stars indicate as p-value, and indicate what the different colours of blue and orange represent. The pictures Fig3c and 3d show different thickness of ONL

RESPONSE #1 We have amended each legend to include these experimental details. See '*black circles denote mean measurement of newly synthesised OS length in each WT and mutant experimental animal; n = 3 animals per genotype; error bars represent standard error of the mean; ***, $p \leq 0.0005$. Different shades of blue and orange represent measurements from individual mice.*'

(#2) Does the intravitreal delivery of the blocking agent influence the function of rhodopsin and therefore vision in the mice?

RESPONSE #2 We have performed ERGs on Rho-SNAP mice following injection of one eye with 0.5 μ l of a SNAP-647 dye into their right eye and no injection into their left eye. There was no significant change in a wave recording on ERG between right and left eyes suggesting there is no gross effect on vision upon ligand treatment. The data has not been added to the manuscript, but we include it as an appendix to this letter (see '**Rebuttal Figure 1. SNAP injection with ERG and OS length**').

(#3) Furthermore, there is missing an essential control in these experiments. Authors should proof that the lengths of outer segments are not changed during the block-chase experiments. Data should be shown after 3-days chase without labelling after 3 days versus 3-days chase with labelling after 3 days. The latter should give no significant differences whereas the first mentioned should. If no differences, the authors should discuss the ongoing phagocytosis by the RPE. Therefore, the conclusion that mutations in Rpgr reduce the rate of photoreceptor disc formation are premature –

RESPONSE #3 We also performed transmission electron microscopy on Rho-SNAP mice following injection of one eye with 0.5 μ l of a SNAP-647 dye into their right eye and no injection into their left eye. We observed no significant difference in outer segment length between the eyes (n = 3 animals per group). The data has not been added to the manuscript, but we include it as an appendix to this letter (see '**Rebuttal Figure 1. SNAP injection with ERG and OS length**').

(#4) Claimed overlap of cofilin with RPGR is not demonstrated by showing RPGR staining in Fig3a and cofilin staining in Fig4d. – line 253-255

RESPONSE #4 We have rephrased the sentence to read '*Using immunofluorescence, we demonstrate that cofilin localises to the photoreceptor CC membrane (Fig. 4d), at the site of disc morphogenesis, similar to RPGR localization (Fig. 3a)*'.

(#5) Fig 4e legend should mention the statistical data. How often have these experiments been performed independently? It seems that there is nonspecific binding of GT335 interactive material, suggesting that the band at ~200kDa (?) is a-specific. In other words, there is no proof for direct binding of cofilin to glutamylated-RPGR. – line 256-258 –

RESPONSE #5 Overexpression of RPGR cDNA is notoriously challenging for several technical reasons, some of which have been bypassed for gene therapy approaches using codon-optimized and shortened constructs.^{1,2} Using both a native full length mouse isoform and a published truncated codon-optimized one (Robin Ali, KCL)¹, as tagged RPGR constructs for mammalian cell overexpression, we have been unable to generate reagents that express RPGR in our hands (see '**Rebuttal Figure 5. Cloning of RPGR and Cofilin plasmids for interaction experiments**'). In contrast, we were able to generate the simpler and shorter tagged cofilin construct. As an alternative approach, we have instead performed a coIP in our *Rpgr*^{Ex3d8} mouse (see new Ext Data Fig. 9f) and show loss of the band in question at ~200 kDa on pull down. Without this additional data, we have therefore rephrased the sentence (now line 268-271) to read '*Co-immunoprecipitation (co-IP) experiments using murine retinal lysates confirmed endogenous cofilin and the retinal specific isoform of RPGR occur in complex in the retina (Fig. 4d).*³³'

(#6) Furthermore, the most essential control is lacking in these experiments (retinal lysates that lack the RPGR protein such as retinal lysates from *Rpgr*Ex3d8)

RESPONSE #6 We thank the reviewer for this suggestion and have repeated the coIP in our *Rpgr*^{Ex3d8} mice and have confirmed loss of the pull down of the band at 220 kDa (see new extended data fig 9f).

(#7) Fig 6 legend should better explain the statistical analysis of these studies depicted in fig6d.

RESPONSE #7 We agree and have added experimental details to all legends, including here where we have edited it to read **(a)** TEM reveals high numbers of vesicles shed from OS bases in *Rpgr*^{Ex3d8} photoreceptors (middle panel) compared to controls (left panel). Shedding of vesicles is reduced upon intravitreal delivery of 25 mM cytochalasin D for 6 hours (right panel) (scale bars = 1 μ m; OS = Outer Segment; * = OS base). **(b)** Quantification of vesicle shedding at the base of photoreceptors by TEM. (Black symbols = mean number of vesicles at the base of each photoreceptor per experimental animal; n = 3 animals per genotype; error bars represent standard error of the mean; *, p \leq 0.05. Different shades of blue, orange and green represent measurements from individual mice). **(c)** TEM reveals intravitreal delivery of 100 μ M LIM kinase inhibitor for 6 hours to wild type retinas leads to elongation of basal discs (blue arrows, left panel) (scale bars = 500 nm). Intravitreal LIMKi delivery to *Rpgr*^{Ex3d8} eyes reduces number of OS shed vesicles (middle and right panels; * = OS base; scale bars = 1 μ m). **(d)** Quantification of vesicle shedding by control or LIMKi treatment from TEM images. (Black symbols = mean number of vesicles at OS base of photoreceptors per experimental animal; n = 3 animals per genotype; error bars represent standard error of the mean; ****, p \leq 0.0001. Different shades of orange and green represent measurements from individual mice).

(#8) The text in the manuscript and the legend to Fig4 do not mention the 4 labelled dysregulated actin binding proteins (two proteins lowered, two proteins increased in levels in the mutant). The mass spec data needs to be deposited in a bank and should become accessible at publication.

RESPONSE #8 We have changed Fig 4a to label the dysregulated actin binding proteins and changed the Fig 4 legend to read: '*(a) Mass spectrometry analysis of Rpgr*^{Ex3d8} retina shows dysregulation of actin binding proteins (PFN1, ARPC3, PDLIM4, PP1 labelled in yellow).' All mass

spec data in the study has been deposited in ProteomeXchange via the PRIDE database. The data is currently private and will be made public on publication of our study.

(#9) The protocol to detect the 6kDa PCD protein should be described or referred to.

RESPONSE #9 The protocol we used to detect the PRCD protein is as described in our methods section under the section 'Protein extraction, antibodies and immunoblotting.'

(#10) The manuscript should more clearly describe that there should be no RPGR protein in the Rpgr^{Ex3d8} strain. Fig3a suggests that there should be no RPGR but there is still a lot of signal which according to the authors is due to aspecific centrosomal staining, without further proof.

RESPONSE #10 To stress to the reader that there should be no RPGR protein in the Rpgr^{Ex3d8} strain, we have altered the legend in Fig 3a to say: '(NB. green staining below polyglutamylation labelling represents non-specific centrosomal staining; a common occurrence with rabbit antibodies. Of note, Ext Data Fig 1b demonstrates the total loss of Rpgr in our Rpgr^{Ex3d8} mouse line, as evidenced by loss on western blot of the 250kDa band on GT335 probing.)'

(#11) In figure 4d, the authors claim that cofilin localises to the connecting cilium, but do not use a ciliary marker to confirm this. The authors should show the proof to this or provide the reference where this colocalization has been shown previously. Also the authors do not show the localisation and presence of cofilin in mutant retina. Is cofilin correctly localised to the CC and at normal levels?

RESPONSE #11 The antibody used for the cofilin staining in Fig 4d was initially provided to us by the Witke lab; their lab is now retired / wound up and were unable to provide more. Instead, we have optimised the use of CST's highly cited monoclonal cofilin antibody (#5175) for use in retinal histochemistry. We show cofilin localising with connecting cilium actin (phalloidin) and, interestingly, cofilin is mislocalised in the Rpgr^{Ex3d8} mouse (Fig 4c). Further, we have rederived the Witke lab's floxed *cofilin* mouse and, with it, created a photoreceptor specific *cofilin* knock-out. We demonstrate specificity of the CC cofilin staining which is lacking in the knock-out. Importantly, by transmission electron microscopy we show that this mouse phenocopies the 'spaced disc' phenotype seen in the Rpgr^{Ex3d8} mouse (Fig 4e).

(#12) -The authors do use the presence of rhodopsin as a measure for raw number of outer segment discs. The authors should provide proof for the correctness of such statement because young mice lacking rhodopsin function temporarily have outer segments with discs. Otherwise, the authors should reference to the literature that photoreceptor discs without rhodopsin do not exist. Or at least discuss that such discs do exist but are less stable.

RESPONSE #12 We appreciate the reviewer's comments in the requirement for rhodopsin in disc generation, but that the *Rho*^{-/-} mouse, whilst grossly having no outer segments, can be found on electron microscopy to have the odd photoreceptor with a shortened OS (Lem., et al, PNAS, 1999). It was therefore crucial to show that our Rhodopsin-SNAP mouse did not impact on photoreceptor function. To this end, we have comprehensively demonstrated in Extended Data Fig 6 that the mice, when maintained heterozygously, have no functional or structural deficit.

(#13) The video (433804_1_supp_7764101_rwndn3) provided that shows SNAP-TMR, SiR-Actin and Hoechst staining, a comparison between wild type and Rpgre^{Ex3d8} is potentially not representative. The wild type does not show a normal layer of ONL, therefore not showing a representative localization of F-actin in the wild type (and overrepresenting the levels of F-actin in the mutant retina).

RESPONSE #13 Live imaging studies of connecting cilium actin dynamics are challenging. Due to the nature of the live tissue preparation, it is difficult to generate optimal retinal slices that are orientated to allow perfect simultaneous visualisation of the whole ONL, connecting cilium and outer segment. On a spinning disc confocal, we are only able to image one z stack to prevent early fluorophore bleaching/phototoxicity and thus allow a 120 minute timelapse. When setting up the experiment, therefore, we choose the plane that best captures the actin labelling in the connecting cilium and, as a result, the ONL is often only depicted by several layers of nuclei. However, it means that CC actin dynamics are best captured and, coupled with the automated analysis, we believe results in an accurate representation of the actin movement within the connecting cilium. This analysis is acquired blind to genotype.

(#14) Did any of the block-chase experiments result in relocalisation of rhodopsin-SNAP in the ONL?

RESPONSE #14 Although we were not specifically looking for this, the retinal analysis we performed in our block chase experiments did not reveal any rhodopsin-SNAP mislocalisation to the ONL.

(#15) Are ectosome vesicles observed as well in RPGR mutant human retina or would the findings be mouse specific?

RESPONSE #15 Because of the limitations of *in vivo* clinical imaging, it is unknown if humans with pathogenic *RPGR* mutations shed vesicles from their outer segment. We predict, however, that our mouse model is representative of human disease and that they do.

(#16) In the proteomics studies there was altered expression of ARP3. Was this tested on Western blots? The authors report a small change in expression of ARP2.

RESPONSE #16 We have tested ARP3 levels with Abcam's monoclonal ARP3 antibody (ab151729) and have detected no significant difference in expression between Rpgre^{Ex3d8} and wild type retinas. We have therefore removed the ARP2 western blot data from the manuscript as we feel it confuses the narrative. Instead, we have focussed on the relationship between RPGR and cofilin, with new *in vivo* data using a photoreceptor specific cofilin knock out mouse.

(#17) Data Fig5d performed blind for genotype ? By at least two independent researchers ? - Data Fig5e seems (overly) convincing but could as well be in part due to inappropriate retinal slice culturing. Blind for genotype, at least 2 researchers. What time of culturing and measurement?

RESPONSE #17 The live imaging protocol was developed in-house using time lapse imaging on the spinning disc confocal Dragonfly imaging platform (90 minutes per time lapse) performed blinded to genotype. Imaging was performed after 2 hours of incubation of slice cultures in fluorescent SNAP

ligands. Protocols for automated quantification of photoreceptor actin dynamics were developed by inhouse image analysis expert (Laura Murphy), again blind to genotype.

(#18) Extended data Fig 3 does not show loss of photoreceptor function. It only shows morphological studies. The extended data Fig 4 does show the loss of retinal function. Change the title of extended data Fig 3.

RESPONSE #18 The title of Ext Data Fig 3 has been changed to '*Extended Data Figure 3. A mouse model of RPGR/XLRP, RpgrORFd5, undergoes loss of photoreceptor structure.*'

(#19) Extended data Fig 6 suggests that the biosensor rhodopsin-SNAP cause decreased outer segment length upon SAP-block

RESPONSE #19 In Extended Data Figure 6f and g, we demonstrate that with no block (i.e. only the SNAP-647 'chase' labelling), the outer segment lengths as detected by SNAP-647 are longer than when the retinas receive the SNAP-647 'chase' labelling 72 hours after a SNAP blocking agent is administered intravitreally. This is therefore proof that the 'chased' SNAP-647 can only bind to the Rhodopsin-SNAP that has been newly generated in the 72 hours since the block agent was administered and thus is only labelling newly generated outer segment. We respectfully contest, therefore, the reviewer's conclusion, that 'the biosensor rhodopsin-SNAP cause decreased outer segment length upon SAP-block. Indeed, we believe we have comprehensively demonstrated this with **RESPONSE #3** above.

(#20) Ext data Fig 6 is unclear. Were the retinal slice in ext data Fig6f naïve retinal slice culture (without SNAP-block) as well cultured for 72 h whereafter stained with fluorescent SNAP ligand? Needs better presentation what the experiment was:

6f (no-block): No intravitreal SNAP-block & SNAP ligand after 72 h culturing

6f (block-chase): Intravitreal SNAP-block & SNAP ligand after 72 h culturing.

Not sufficient clear that this is consistent with newly formed discs. It might just interfere with rhodopsin function. In this experiment with wild type mice it has nothing to do with RPGR?

RESPONSE #20 The reviewer is correct that the experiments in Ext Data Fig 6 have nothing to do with RPGR but were performed on 'wild type' rhodopsin-SNAP mice. We believe the ERG data showing normal scotopic and photopic retina function confirms that the SNAP tag does not affect rhodopsin function.

We thank the reviewer for their suggestion to improve the clarity of the figure and have amended the panels in Ext data fig 6f to say:

Left panel: No block (just chase)

Right panel: Block-chase'

(#21) - Seems as if f-actin is present in the ectosomes?

RESPONSE #21 This is an interesting question and one that was partially answered in a different mouse model by Spencer et al (2019),³ who showed by EV proteomics that both actin and many actin binding proteins to be present within the vesicles shed from the *rds/peripherin* mouse. We would therefore expect the vesicles shed from our *Rpgr* mutant mice to be similar in composition. However, when the reviewer mentions that it 'seems as if F-actin is present in the ectosomes' in our work, we believe they are referring to the image in Fig 5b. We stress that the circular, ectosome-looking structures are in fact nascent discs, as can be seen when a movie of the whole tomogram is played (see new 'Extended Data Movie 1' submitted as part of this revision).

(#22) Knockout of RPGR still actin binding protein activity >> slows down how much?

RESPONSE #22 This is a fascinating question. We agree that perturbation of RPGR likely only slows down the activity of actin severing proteins at the site of disc formation by compromising their activity, rather than stopping it entirely. This would be in keeping with the slow nature of the photoreceptor degeneration seen in both our mouse models and human patients. How much it slows down their activity is beyond the scope of this paper, but we at least have an idea from our pulse chase experiments how much the rate of disc formation is slowed.

Reviewer #2:

(#1) Ln 196 Fig 3a was confusing. Indications, in the images, of the location of CC and centrosome would be helpful. Also the non-specific staining using RPGR antibody seems different in the wild type and the mutant. Perhaps an extended figure showing the non-specificity would be helpful to show that the difference is purely a technical issue.

RESPONSE #1 To Fig 3a, we have added annotations to indicate the CC and the level of the non-specific centrosomal staining. We hope that, by better directing the reader to this non-specific binding (of note Ext Data Fig 1b confirms total loss of RPGR's retinal specific protein by western blot), the reviewer can appreciate this non-specific staining looks similar between wild type and mutant. Unfortunately, the scientist in the Wensel lab (Houston) with the STORM expertise and who compiled Fig3a has moved to a lab in Europe. Thus, generating further figures is difficult, but we hope that these additions will serve to show that the difference is purely a technical issue.

(#2) Ln 292 Please identify if "n" is animals or something else.

RESPONSE #2 We have amended the Figure 5 legend to denote that n refers to animals.

(#3) Ln 315 Please include procedure for LIMKi use in the methods. Also it would be helpful for comparison with other past or future work and for possible translational studies to estimate what the concentration of LIMKi and cytochalasin is in the mouse eye.

RESPONSE #3 The following has been added to the methods: *'LIMKi rescue experiment: Mice were anesthetised using isoflurane and pupils were dilated through the topical application of 1% w/v tropicamide. 0.5 µl of 100µM LIMK inhibitor (TOCRIS CRT 0105950) in PBS was injected intravitreally. Contralateral eyes had a sham injection of 0.5 µl PBS and served as the control eye. 6 hours later mice underwent transcardial perfusion using fixative and processed for electron microscopy as outlined above.'*

(#4) Ln 482 & 568 The abbreviation "OCT" is used for two different things- optical coherence tomography and optimal cutting temperature. Please change the abbreviation for one of these.

RESPONSE #4 OCT is no longer used as an abbreviation for optimal cutting temperature. Indeed, this has also been removed as an abbreviation on line 550

(#5) Ln 550 Change "to" to "for"

RESPONSE #5 This has been altered.

(#6) Ln 554-557 Please rewrite this sentence. Does not make sense.

RESPONSE #6 This sentence has been changed to: 'For wax preservation, eyes were fixed in Davidsons fixative overnight at 4 °C. Following fixation in Davidsons fixative, eyes were incubated

successively in 70% v/v, 80% v/v, 90% v/v and 100% v/v ethanol, twice in xylene and then paraffin, each for 45 min per stage, using a vacuum infiltration processor.

(#7) Ln 598 I think “Grids” should say “sections” for more accuracy.

RESPONSE #7 This has been altered.

(#8) Ln 603 Please add “were” processed.

RESPONSE #8 This has been added.

(#9) Ln 649-650 Please remove italics.

RESPONSE #9 These has been removed.

(#10) Ln 690 Remove the word “frozen”.

RESPONSE #10 This has been removed.

(#11) Ln 706 A space might be added between (bin4) and voxels.

RESPONSE #11 This has been done.

(#12) In general, the methods section should be carefully reviewed as this reviewer only looked at methods that were known to them.

RESPONSE #12 We have carefully reviewed the methods section and corrected all errors we could find.

Reviewer #3:

(#1) In a key aspect of this manuscript, the authors suggest that RPGR interacts with cofilin based on co-immunoprecipitation and co-localization results. However, the data is not overly convincing as it is based on a single coimmunoprecipitation result shown in Figure 4e. The blot itself is of relatively poor quality and co-immunoprecipitation does not necessarily prove that there is a direct interaction of RPGR and cofilin. If one were to believe this co-immunoprecipitation experiment, then it is possible that the cofilin antibody is pulling down a large complex containing RPGR, but not necessarily binding directly to RPGR. More direct interaction and the effect of RPGR in cofilin activity is needed to draw a conclusion regarding the interaction of cofilin and RPGR and the effect of RPGR on cofilin actin depolymerizing activity. The manuscript would be strengthened if the authors could isolate RPGR and cofilin and show that they directly interact and that RPGR modulates the depolymerization activity of cofilin.

RESPONSE #1 We agree with the reviewer and have amended the manuscript to state that our experiments provide evidence that the retinal-specific isoform of RPGR occurs in complex with cofilin. We have also provided several additional experiments supporting this interaction and the effect that RPGR has on cofilin activity (See lines 261-275 and Fig. 4). Firstly, immunofluorescence studies show that cofilin's localisation to the connecting cilium is lost in the *Rpgr^{Ex3d8}* mouse. Secondly, we engineered a photoreceptor-specific cofilin knock-out mouse whereby TEM we show that it phenocopies the 'spaced disc' phenotype seen in the *Rpgr^{Ex3d8}* mouse. Whilst we also attempted to generate tagged cofilin and RPGR overexpression constructs for pull down studies, we have been unable to generate expression vectors for *Rpgr* in the time frame of this revision (see '**Rebuttal Figure 5. Cloning of RPGR and Cofilin plasmids for interaction experiments**'). This likely reflects its repetitive open reading frame, even when using the truncated codon-optimised sequence used by Robin Ali (KCL)¹ for a gene therapy approach. Nonetheless, we feel that these other new data support that RPGR acts to regulate cofilin activity in the photoreceptor to regulate disc formation.

(#2) Figure 2: It would help the reader if a higher magnification image were presented to show the differences in disc compaction and spacing by TEM.

RESPONSE #2 As requested, we have included in Figure 2 higher magnification images depicting the changes in disc compaction and spacing.

(#3) It is unclear why the authors use PRCD as an indicator to show that rod outer segments of *Rpgr(Ex3d8)* retinas contain lower photoreceptor disc component than WT retina. Proteomic data should show a decrease in all the outer segment specific membrane proteins including rhodopsin, peripherin2, ABCA4, etc. This should be added to the manuscript to provide the reader with confirmation on the difference in length of the outer segments.

RESPONSE #3 We were interested in the reduction in PRCD in our mass spec data because it was significantly down in both our datasets, which were generated from mice at different ages and performed at different facilities. It was of particular interest because knock-out of the gene in mice also leads to abnormal outer segments and shed vesicles. Our mass spec data did not show an equal decrease in rhodopsin, peripherin2 and ABCA4. Confirmation of PRCD's reduction by western blotting

was in keeping with our measurements of mice outer segment length by transmission electron microscopy, that specifically shows shortened outer segments when RPGR is perturbed. Further, our human data shows shortened outer segments when RPGR is perturbed. We have previously shown that perturbing RPGR results in rhodopsin mislocalisation to the photoreceptor perinuclear area (possibly the endoplasmic reticulum) and the outer plexiform layer (Megaw et al., 2017). Therefore, we would not expect a reduction in photoreceptor rhodopsin levels, even though outer segments are shortened. We attempted to quantify the amount of ABCA4 in our retinal lysates by western blotting (Antibodies online AA2250-2263), but were unable to identify a band at the expected size on the blot (**Rebuttal Figure 4. ABCA4 western**). We could not identify a commercial antibody for peripherin2 and were unable to get our hands on the previously generated in house antibody. We have therefore amended line 175-7 to say, '*whilst no differences were observed on mass spec of other outer segment proteins (e.g. peripherin2, ABCA4), reduced PRCD in Rpgg^{Ex3d8} retinas was confirmed on immunoblotting (Fig. 2e,f).*'

(#4) Figure 4e: There are many bands in the immunoblot of RPGR. The authors should inform the reader what these bands represent. If they are crossreacting proteins, then this could comprise the studies involving immunofluorescence microscopy and would suggest that the antibody are not overly specific. – cofilin immuno of the flox-cof

RESPONSE #4 We have repeated the CoIP experiment using the *Rpgg* knock out mouse (see new Ext Data Fig. 9f). This shows that the band of interest at 220 kDa is gone but that the bands at 50 kDa and 25 kDa remain, which we believe to be heavy (50 kDa) and light (25 kDa) immunoglobulins sticking to the magnetic beads during CoIP. We also know that there is an expected band representing tubulin at 50 kDa, which is polyglutamylated and therefore labelled by the GT335 antibody. We have altered the figure legend to better inform the reader.

(#5) As indicated above, it would be important to show a direct interaction or regulation of cofilin by RPGR such as using heterologous expression system together with co-IP and activity assays.

RESPONSE #5 As mentioned above in **RESPONSE #1**, we attempted to generate tagged cofilin and RPGR constructs for overexpression/pull down studies. We were able to generate the tagged cofilin construct. However, due to RPGR's repetitive open reading frame and large size, we were unable to generate a tagged RPGR construct, despite using the truncated sequence optimised by Robin Ali (KCL) for a gene therapy approach (see '**Rebuttal Figure 5. Cloning of RPGR and Cofilin plasmids for interaction experiments**'). We acknowledge, therefore, that these studies do not confirm direct interaction between the two proteins and so have rephrased the sentence to read '*Co-immunoprecipitation (co-IP) experiments using murine retinal lysates confirmed endogenous cofilin and the retinal specific isoform of RPGR occur in complex in the retina (Fig. 4d).*^{33'} (Line 269-71)

(#6) In the discussion, it is unclear how the interaction of RPGR with gelsolin relates to the potential interaction of RPGR with cofilin. It would be helpful if the authors discuss how RPGR may regulate actin dynamics via these actin severing proteins.

RESPONSE #6 We believe that RPGR acts as a scaffold protein in the connecting cilium, bringing the relevant actin severing proteins (e.g. cofilin and gelsolin) into contact with the actin microfilaments

involved in basal disc morphogenesis, allowing for their timely disassembly. We have stated this in sentence three of our discussion.

Reviewer #4:

(#1) My main concern of this study is the preservation of the samples used for electron microscopy and tomography. Many of these samples appear suboptimal or it is difficult for the reader to determine the level of preservation based on how the data is presented. This is particularly relevant for the tomography data as a single slice from each tomogram is shown of a small area of the photoreceptor. Supplementary videos of the tomography data should be included so that the readers can see each slice from the full data stack.

RESPONSE #1 As requested, we have provided supplementary movies of the tomography data, namely those from Fig 5a and b (see Ext Data Movie 1 and 2). These allow the reader to view every slice through the tomogram stack and therefore, we hope, reassure the reader as to the quality of the tomograms.

(#2) It would help to have lower magnification cryo electron microscopy images in the main manuscript or supplementary data to show more photoreceptors and provide evidence that they are preserved well enough to be assessed by cryo electron tomography. The reviewer acknowledges that preserving photoreceptors for cryo-electron microscopy is difficult and there are limitations, but it is important to be sure that the photoreceptors are not damaged, and the membrane have not been distorted and actin filaments displaced.

RESPONSE #2 We have provided a supplementary figure depicting lower power cryo EM images of mutant and wild type photoreceptors to demonstrate that the technique we use, as optimised by the Wensel lab (Gilliam et al., 2012)⁴, where unfixed samples are flash frozen to preserve them in their native aqueous environment, preserves photoreceptors (See new Ext Data Fig. 1). We stress that our preparation technique results in isolated cell fragments and are not expected to be completely “intact” as, during isolation, membranes can break and reseal, the stack of basal discs can detach from its connections to the fully formed discs, and the cells tend to flatten to some extent on the grid. However, as we (Gilliam et al., 2012; Dharmat et al., 2018; Robichaux et al., 2019) and others (Palczewski & Baumeister laboratories) have found consistently, internal structures such as filaments and microtubules are well preserved (although their exact geometrical relationships to one another may not be), and we would contend that fewer “distortions” are observed than those associated with conventional TEM, due to fixation, dehydration, plastic embedding, heavy metal staining, etc.

(#3) In Figure 1 the tomography data shows an odd-looking structure (due to its size, shape and contents) described as basal discs. From the single slice in (b) there appears to be membranes that are not included in the segmentation. Due to this and potentially other structures being excluded, the segmentation does not seem to match up well with the tomography slice. Either more of the membrane should be segmented to reflect the tomography data or there needs to be more clarity about what has been included or excluded from the segmentation in (b).

RESPONSE #3 We have replaced the segmented image in Fig 1c such that it more closely matches the tomography slice in 1b. For this experiment, we aimed to analyse the newest, most basal disc emerging from the base of the outer segment, reasoning that it would be the disc most likely to shed light on the mechanism of disc formation. Thus, the membranes above the basal disc, visible in Fig

1b, were not included for the lengthy analysis process. We have included the following in the manuscript (Lines 103-105) to explain this: 'Flash freezing isolated mouse rod photoreceptor outer segments (ROS) allows visualisation of 3-dimensional architecture at an ultrastructural level by creating 3-dimensional maps from a tilt series of electron tomograph images. To best distinguish between an active and a passive process, our annotation focussed on the nascent disc emerging from the base of the CC.'

In segmenting the structure labelled as "basal disks (BD)" in the tomogram of Fig. 1., we selected the outermost membrane and the filaments that could be identified within it. There were additional membrane structures within it as well, but segmenting these would obscure the view of the filaments which *cannot* be easily seen in the slice projection of the raw tomogram. We have substituted in place of the original map projection of 1b, a projection image of a section within the map that more closely matches the segmented view shown.

(#4) In Figure 1 panels (e) and (f), it is difficult to compare the subtomographic reconstruction against the previously published actin structure. It would be helpful to have movies to show these filaments rotating around the y-axis.

RESPONSE #4 We have provided a supplementary movie showing the subtomographic reconstructions rotating around the y axis with cofilactin lying over the wild-type subtomographic reconstruction and actin lying over the *Rpgr^{Ex3dB}* subtomographic reconstruction (see new Ext Dat Movie 3).

(#5) Extended Data Movie 1 shows some slight actin movement, but this could be the actin within the connecting cilium, rather than the basal discs. The photoreceptor also appears to be moving too, so it is hard to draw conclusions about the actin dynamics from this data.

RESPONSE #5 We agree that the actin movement depicted using spinning disc microscopy of our retinal slice cultures does not offer the resolution to conclude whether the actin is within the connecting cilium or the basal disc. To reflect this, the manuscript (lines 301-302) reads: 'Analysis showed reduced actin dynamics in the CC of *Rpgr^{Ex3dB}* photoreceptors (Fig. 5d, e, Extended Data Movies. 1 and 2).' This reduced actin dynamics, observed in *Rpgr* mutant mice is still relevant, given our proposed model that RPGR regulates actin turnover in the photoreceptor CC.

All efforts were made to immobilise the slice culture retinas prior to analysis. This included using an autoregressive motion tool in the Imaris software, which stabilised translational movement of photoreceptor nuclei and applied this stabilisation to the SiR-actin-labelled structures.

(#6) Based on the results shown in Figure 1 there is not enough evidence to say 'We thus conclude that dynamic actin changes occur within nascent photoreceptor discs, suggesting a role for actin in disc genesis'. As it is not clearer that the structure containing actin are basal discs and actin polymerisation can not be clearer assessed from the time lapse videos.

RESPONSE #6 We agree and have altered the manuscript (Line 113-114) to read: 'We thus conclude that dynamic actin changes occur within the photoreceptor CC, supporting a role for actin in disc genesis.'

(#7) In Figure 2 (a) the wild-type panel is not a good representation of the OS length as photoreceptor are slightly oblique and entire OS are not shown. Any obliqueness of the OS would impact on the measurements made in (b). It would also be better to show images that include the edge of the RPE, to clearly show the full length of the OS and that they are shorter in the *Rpgr*^{EX3d8} model.

RESPONSE #7 We have added a further figure, 'new Extended Data Figure 6', which contains non oblique images of the outer segment, as well as the edge of the RPE and thus better demonstrates the shortened outer segments of *Rpgr*-mutant photoreceptors. We have kept the images in Fig 2a as we feel it offers a good representation of the split disc phenotype seen in our *Rpgr*-mutant photoreceptors, as well as highlighting the vesicles observed throughout the *Rpgr*-mutant photoreceptor layer.

(#8) The authors should be careful about using the term 'split disc' phenotype, as this can be a sample preparation artefact rather than a genuine phenotype. An example of this is in Figure 6 (a) for the WT untreated sample there are greater gaps between some of the discs that are not as evident in the better-preserved WT sample in Figure 2 (c).

RESPONSE #8 The reviewer correctly raises the concern that poorly fixed TEM sample can result in poor preservation of photoreceptor outer segment architecture; indeed, the separation of discs can result. Instead, we have replaced the term with 'spaced disc' phenotype. We stress, however, that our lab uses the preparation technique pioneered by the Arshavsky lab, that is accepted as the gold standard for TEM preparation. Further, all mutant and wild type mice for these experiments were prepared in the same way and acquisition of images is blinded as to genotype. In short, the 'spaced disc' phenotype is genuine, reproducible and robust on TEM imaging for now two mouse mutants *Rpgr* and *Cofilin*. To convince the reviewer, we have carried out quantification of the extent of the disc splitting within a random field across the outer segments. In ImageJ, we inverted each TEM image and made binary calls of each pixel in the images (i.e. we asked ImageJ to call the pixel as either black or white). We then made measurements of the pixel intensity along a single line spanning, as best possible, the length of an outer segment. The mean pixel intensity of each line was then calculated, and showed a significant reduction in mutant mice, signifying the spaces in the mutant mice were significantly more than in wild type mice. We have chosen not to include the data in the revised manuscript, as we feel it unnecessary, but include it as an appendix to this letter (see **Rebuttal Figure 4. Quantification of split disc phenotype**).

(#9) When examining the images in Figure 2 (a) and (b) the OS could potentially be wider in *Rpgr*^{EX3d8} model. It would be good to check to be certain that only the length is affected.

RESPONSE #9 As requested, we have measured outer segment disc width as evidenced by transmission electron microscopy imaging (n = 3 - 4 animals per genotype). As the reviewer correctly postulated, the discs appear to be wider in the *Rpgr*^{EX3d8} mutant mouse. This would be entirely in keeping with a slowed rate of disc formation, which would result in excess membrane being added to nascent discs as the actin-mediated disc completion is slowed. We have included the measurements as an appendix to this letter (see '**Rebuttal Figure 2. WT v Ex disc widths**'). However, we have elected not to include it in the main manuscript as we are concerned that disc width measurements are susceptible to over-/underestimation if the plane of the TEM image does not capture the full width of the disc. The measurements will be available for the reader through this rebuttal letter, but we feel it

best to withhold it from the main paper.

(#10) In the text or in the legend of Figure 3, there needs to be more information about the polyglutamylation staining to say that is used to label polyglutamylated tubulin. Furthermore, is the polyglutamylation staining only in the connecting cilium or does it extend into the OS? From the images in Figure 3a it seems that RPGR does not extend as far as the polyglutamylation staining and therefore, may not extend all the way up the connecting cilium to the site of disc formation.

RESPONSE #10 Polyglutamylated tubulin extends the length of the photoreceptor connecting cilium and into the outer segment. We have therefore amended Fig 3's legend, which now states: '(a) Top panel: Localisation of RPGR's retinal-specific isoform extends the length of the photoreceptor connecting cilium, as evidenced by SIM imaging, showing co-localisation with polyglutamylated tubulin in wild-type photoreceptors, which is known to extend throughout the CC and into the OS.'

(#11) If actin filaments extend into newly forming discs are these released in ectosome in the RpgrEX3d8 model? Can you see filaments or other structures in the ectosomes when examined by cryo-electron tomography (Figure 5 (b)), or by room temperature electron microscopy or tomography.

RESPONSE #11 This is a super interesting question and one that was partially answered in a different mouse model by Spencer et al (2019)³, who showed actin and many actin binding proteins to be present within the vesicles shed from the *rds/Peripherin* mouse. We would therefore expect the vesicles shed from our *Rpgr* mutant mice to have a similar composition but have not undertaken any EV proteomic studies to date to confirm. The reviewer specifically asks about Fig5b. We stress that these circular, ectosome-looking structures seen in this single tomographic slice are, in fact, basal discs, as can be seen when a movie of the whole tomogram is played (see new Extended data movie 2 submitted as part of this revision in response to this reviewer's previous comment). Our room temperature electron microscopy data does not offer the resolution to determine whether actin is present within the shed vesicles.

(#12) When looking at Extended Figure 7 the OS of the RpgrEX3d8 model appear to be longer than the WT. The WT is perhaps slightly oblique but according to Figure 2b the OS should be approximately 25% shortening of the OS in the RpgrEX3d8 model, which not evident in this data. –

RESPONSE #12 We thank the reviewer for this observation and have replaced the wild-type panel in question with a more representative, non-oblique image (now in new Extended Figure 9), which supports the outer segment length measurements in Fig 2b.

(#13) In Figure 4 the measurements from the Western blots in (b) and (c) are not convincing. In (b) the α -tubulin looks over exposed and there is no visible difference between the bands for ARP2 and phospho Cofilin. This is unlike Figure 2 (e) and (f) where you have a clear visible difference in the PRCD bands. – Redo

RESPONSE #13 We have repeated the cofilin western, now shown in Fig 4b, which confirms the increased phosphorylation at serine 3 in our mutant retinas. We believe the new blot better shows the stronger pCofilin band in the mutant lanes, but stress that the phospho Cof : total Cof ratio, calculated using intensity measurements of blot bands, repeatedly demonstrates this using 5-7 biological

samples per genotype. We have also performed western blotting using an ARP3 antibody (ab151729), which showed no change between mutant and wild type retinas and have thus removed the ARP2 blot from our manuscript. This helps focus the narrative on cofilin's role in the photoreceptor, with the addition of significant new in vivo experimental data using a cofilin knock out mouse model (see new Fig. 4e).

(#14) As a comparison to the Cofilin staining in Figure 4 (d) it would be good to show the staining in the Rpgr^{EX3d8} model. Co-staining of RPGR and Cofilin should be included to show colocalization between the two. It would also add to the paper to show co-staining of actin and Cofilin to see if there is complete colocalization of the two within the connecting cilium and base of the OS.

RESPONSE #14 The antibody used for the cofilin staining in Fig 4d was initially provided to us by the Witke lab; their lab is now retired / wound up and were unable to provide more. As a commercially available alternate, we have optimised the use of CST's highly publicized monoclonal cofilin antibody for use in retinal histochemistry. We show cofilin localising with connecting cilium actin (phalloidin) and, interestingly, cofilin mislocalization in the Rpgr^{EX3d8} mouse (new Fig 4c). In addition, we have rederived the Witke lab's floxed *Cofilin* mouse model and, with it, created a photoreceptor specific cofilin knock out. Here, we demonstrate that the connecting cilium cofilin staining is lacking, attesting to the specificity of the reagent/signal. Moreover, we show by transmission electron microscopy studies show that conditional retina cofilin KO phenocopies the 'spaced disc' phenotype as seen in the Rpgr^{EX3d8} mouse, at a similar early age (new Fig 4e).

(#15) As mentioned above for Figure 5 it is important to include supplementary videos of the tomograms and provide more images to show that the samples have been adequately preserved and to allow the reader to see the length of the actin filaments.

RESPONSE #15 As above, we have now included Extended Data Movies of the tomograms to show sample preservation (see new Extended Data Movies 1 to 3).

(#16) In extended movie 2 the preservation of the wild type is very poor compared to Rpgr^{EX3d8}. If this was used for the analysis in figure 5 (d), the preservation is too poor to be used. Furthermore, there seems to be very little actin movement in the movie, with most actin movement occurring within inner segment region.

RESPONSE #16 Live imaging studies of connecting cilium actin dynamics are challenging. Due to the nature of the live tissue preparation, it is difficult to generate live retinal slices that are orientated to allow perfect visualisation of the whole ONL, connecting cilium and outer segment at the same time. Using a spinning disc confocal, we are only able to image one z stack to prevent early fluorophore bleaching and phototoxicity, thus allowing a 120 minute timelapse. When setting up the experiment, therefore, we choose the Z-plane that best captures the actin labelling in the connecting cilium and, as a result, the ONL is often only depicted by several layers of nuclei. However, it means that CC actin dynamics are best captured and, coupled with the automated analysis, we believe results in an accurate representation of the actin movement within the connecting cilium. The reviewer correctly points out that there is relatively little actin movement in the movie with this resolution. However, over the course of the timelapse movie, it is able to accurately measure it. We are excited to be able to

apply emerging technologies like lattice lightsheet to extend our resolution and limit phototoxicity/bleaching to allow longer imaging windows in the near future.

(#17) In the text there is mention that there is disc overgrowth when treating mice with cytochalasin D, but this is not evident in Figure 6. An image of WT mouse treated with cytochalasin D should be included so that it can be compared to the RpgrEX3d8 model.

RESPONSE #17 We have provided an additional figure, new **Extended data figure 11**, that demonstrates overgrowth of the basal discs when wild-type mice are treated with cytochalasin D.

(#18) In Extended Data Figure 8, the correlation coefficient calculated when comparing the wild type and RpgrEX3d8 subtomographic averages is not very meaningful, considering part of the wild type reconstruction is poorly resolved and most of the overlapping structure is the central region of the filaments. The wild type structures with and without imposed helical symmetry looks different from the structures shown in Figure 1. Where these generate from a different set of tomograms? –

RESPONSE #18 We have removed the correlation coefficient data from our manuscript.

References

1. Pawlyk BS, Bulgakov OV, Sun X, Adamian M, Shu X, Smith AJ, Berson EL, Ali RR, Khani S, Wright AF, Sandberg MA, Li T. Photoreceptor rescue by an abbreviated human RPGR gene in a murine model of X-linked retinitis pigmentosa. *Gene Ther.* 2016 Feb;23(2):196-204. doi: 10.1038/gt.2015.93. Epub 2015 Sep 8. PMID: 26348595; PMCID: PMC4863462
2. Fischer MD, McClements ME, Martinez-Fernandez de la Camara C, Bellingrath JS, Dauletbekov D, Ramsden SC, Hickey DG, Barnard AR, MacLaren RE. Codon-Optimized RPGR Improves Stability and Efficacy of AAV8 Gene Therapy in Two Mouse Models of X-Linked Retinitis Pigmentosa. *Mol Ther.* 2017 Aug 2;25(8):1854-1865. doi: 10.1016/j.ymthe.2017.05.005. Epub 2017 May 24. PMID: 28549772; PMCID: PMC5542800.
3. Spencer WJ, Lewis TR, Phan S, Cady MA, Serebrovskaya EO, Schneider NF, Kim KY, Cameron LA, Skiba NP, Ellisman MH, Arshavsky VY. Photoreceptor disc membranes are formed through an Arp2/3-dependent lamellipodium-like mechanism. *Proc Natl Acad Sci U S A.* 2019 Dec 26;116(52):27043-27052. doi: 10.1073/pnas.1913518117. Epub 2019 Dec 16. PMID: 31843915; PMCID: PMC6936530.
4. Gilliam JC, Chang JT, Sandoval IM, Zhang Y, Li T, Pittler SJ, Chiu W, Wensel TG. Three-dimensional architecture of the rod sensory cilium and its disruption in retinal neurodegeneration. *Cell.* 2012 Nov 21;151(5):1029-41. doi: 10.1016/j.cell.2012.10.038. PMID: 23178122; PMCID: PMC3582337.
5. Dharmat R, Eblimit A, Robichaux MA, Zhang Z, Nguyen TT, Jung SY, He F, Jain A, Li Y, Qin J, Overbeek P, Roepman R, Mardon G, Wensel TG, Chen R. SPATA7 maintains a novel photoreceptor-specific zone in the distal connecting cilium. *J Cell Biol.* 2018 Aug 6;217(8):2851-2865. doi: 10.1083/jcb.201712117. Epub 2018 Jun 13. PMID: 29899041; PMCID: PMC6080925.
6. Robichaux MA, Potter VL, Zhang Z, He F, Liu J, Schmid MF, Wensel TG. Defining the layers of a sensory cilium with STORM and cryoelectron nanoscopy. *Proc Natl Acad Sci U S A.* 2019 Nov 19;116(47):23562-23572. doi: 10.1073/pnas.1902003116. Epub 2019 Nov 5. PMID: 31690665; PMCID: PMC6876244.

REVIEWER COMMENTS

Reviewer #1 (Remarks to the Author):

The authors sufficiently addressed the questions raised in the revised manuscript.

Reviewer #3 (Remarks to the Author):

The authors have made substantial changes as requested by this and other reviewers. Most issues have been resolved. The results contribute to our understanding of photoreceptor outer segment morphogenesis and the role that RPGR plays in this process and X-linked RP

Reviewer #4 (Remarks to the Author):

The authors have done considerable work to address the reviewers' comments. I still have some concerns regarding the sample preservation for the cryo electron tomography.

In Extended Data Figure 1 are all the images of wild-type mouse photoreceptors? From this new supplementary data that you have included, is evident that the basal outer segment is not well preserved. It is likely that the actin is preserved, but it is certainly possible that it has been displaced and the basal discs are damaged. The damage will have likely occurred when isolating the photoreceptors and is mostly unavoidable. It is good that the readers can now get a better overview of the samples that were analysed for the tomography from the supplementary images and videos that have been added to the manuscript. It is important to include more information about the limitations of this method within the manuscript similar to what you described in your rebuttal – “during isolation, membranes can break and reseal, the stack of basal discs can detach from its connections to the fully formed discs, and the cells tend to flatten to some extent on the grid. However, as we (Gilliam et al., 2012; Dharmat et al., 2018; Robichaux et al., 2019) and others (Palczewski & Baumeister laboratories) have found consistently, internal structures such as filaments and microtubules are well preserved (although their exact geometrical relationships to one another may not be)”.

It is good to see that the reconstruction in figure 1c now matches the image on the tomography data in figure 1b. It seems that the region labelled BD are not the most basally positioned or the newest discs. There seems to be a few discs with narrow spacing in-between that are positioned below the structure labelled BD. The BD structure is also very large and reflects damage to the basal outer segment, which

gives further reason to explain the limitations of the sample preparation and the type artifacts that can arise. If you are only rendering some of the membranes within the BD region of the reconstruction this needs to be mentioned in the manuscript so that this is clear to the reader.

A video should be included for the tomogram in figure 1 and the resolution of this and the other two videos should be improved to make them high enough to make out more features. Currently, the resolution/detail is very low, and when looking at regions that are supposed to have the actin filaments, they are grainy, and it is not possible to make out any structure. The authors should also consider uploading the tomograms to an online repository such as Electron Microscopy Data Bank.

The wild-type actin structure from the subtomographic averaging in Extended Data Figure 10 looks different from the structure shown in Figure 1. Where these generate from a different set of tomograms?

Response to reviewer #4 comments:

(#1) In Extended Data Figure 1 are all the images of wild-type mouse photoreceptors? From this new supplementary data that you have included, is evident that the basal outer segment is not well preserved. It is likely that the actin is preserved, but it is certainly possible that it has been displaced and the basal discs are damaged. The damage will have likely occurred when isolating the photoreceptors and is mostly unavoidable. It is good that the readers can now get a better overview of the samples that were analysed for the tomography from the supplementary images and videos that have been added to the manuscript. It is important to include more information about the limitations of this method within the manuscript similar to what you described in your rebuttal – “during isolation, membranes can break and reseal, the stack of basal discs can detach from its connections to the fully formed discs, and the cells tend to flatten to some extent on the grid. However, as we (Gilliam et al., 2012; Dharmat et al., 2018; Robichaux et al., 2019) and others (Palczewski & Baumeister laboratories) have found consistently, internal structures such as filaments and microtubules are well preserved (although their exact geometrical relationships to one another may not be)”.

RESPONSE #1 The reviewer is correct in that extended Data Figure 1 are all images of wild-type mouse photoreceptors. As requested, we have added the following to the results section in the manuscript (lines 123-5): ‘Although membranes can break and reseal during the isolation process, internal structures such as filaments are well preserved. (Dharmat et al. 2018; Gilliam et al. 2012)’

(#2) It is good to see that the reconstruction in figure 1c now matches the image on the tomography data in figure 1b. It seems that the region labelled BD are not the most basally positioned or the newest discs. There seems to be a few discs with narrow spacing in-between that are positioned below the structure labelled BD. The BD structure is also very large and reflects damage to the basal outer segment, which gives further reason to explain the limitations of the sample preparation and the type artifacts that can arise. If you are only rendering some of the membranes within the BD region of the reconstruction this needs to be mentioned in the manuscript so that this is clear to the reader.

RESPONSE #2 We have added to the manuscript (lines 125-6): ‘To best distinguish between an active and a passive process, our annotation focussed on the unflattened, nascent disc emerging from the base of the CC.’

(#3) A video should be included for the tomogram in figure 1 and the resolution of this and the other two videos should be improved to make them high enough to make out more features. Currently, the resolution/detail is very low, and when looking at regions that are supposed to have the actin filaments, they are grainy, and it is not possible to make out any structure. The authors should also consider uploading the tomograms to an online repository such as Electron Microscopy Data Bank.

RESPONSE #3 We have included this movie, as requested. We regret, however, that the raw tomogram movies are low resolution because the data are low resolution, which is an intrinsic feature of cryo-ET and why we perform sub-tomogram averaging. There is sadly nothing that can be done about this. We will also deposit the data in the EMDB database, as we have done for previous cryo-ET publications.

(#4) The wild-type actin structure from the subtomographic averaging in Extended Data Figure 10 looks different from the structure shown in Figure 1. Where these generate from a different set of tomograms?

RESPONSE #4 The subtomogram average in Fig. 1 was an average from tomograms of both WT and *Rpgr*^{Ex3d8} mice. Our initial goal was to confirm that the objects we were studying were, indeed, actin filaments. As individual filaments are low-contrast and very noisy, we used as many as we could find to facilitate comparison to the previously published high-resolution structure.

Later, we sought to compare such reconstructions between the two genotypes, and from that comparison, extended figure 10 was produced (now extended figure 8 in revised manuscript).

REVIEWERS' COMMENTS

Reviewer #4 (Remarks to the Author):

I am satisfied that the authors have addressed my comments.